# Radiative cooling and indoor light management enabled by a transparent and self-cleaning polymer-based metamaterial

Gan Huang [1] ✉, Ashok R. Yengannagari[1], Kishin Matsumori [1], Prit Patel[1], Anurag Datla[1], Karina Trindade[1], Enkhlen Amarsanaa[1], Tonghan Zhao [1], Uwe Köhler[1], Dmitry Busko[1] & Bryce S. Richards [1,2] ✉

Transparent roofs and walls offer a compelling solution for harnessing natural light. However, traditional glass roofs and walls face challenges such as glare, privacy concerns, and overheating issues. In this study, we present a polymer-based micro-photonic multi-functional metamaterial. The metamaterial diffuses 73% of incident sunlight, creating a more comfortable and private indoor environment. The visible spectral transmittance of the metamaterial (95%) surpasses that of traditional glass (91%). Furthermore, the metamaterial is estimated to enhance photosynthesis efficiency by ~9% compared to glass roofs. With a high emissivity (~0.98) close to that of a mid-infrared black body, the metamaterial is estimated to have a cooling capacity of ~97 W/m² at ambient temperature. The metamaterial was about 6 °C cooler than the ambient temperature in humid Karlsruhe. The metamaterial exhibits super-hydrophobic performance with a contact angle of 152°, significantly higher than that of glass (26°), thus potentially having excellent self-cleaning properties.

The utilization of transparent roofs and walls in buildings has gained significant attention as an effective means of harnessing natural light, reducing energy consumption[1,2], and improving occupants' well-being[3,4]. The integration of transparent roofs and walls in architectural design offers numerous advantages, such as maximizing daylight utilization and creating visually appealing spaces. Natural light has been shown to positively impact human health and productivity[5], leading to improved comfort and mood. In addition, transparent roofs and walls contribute to energy efficiency by reducing the need for artificial lighting[6], thus playing a crucial role in achieving sustainable building certifications and meeting stringent energy performance standards. Green building rating systems, such as leadership in energy and environmental design, prioritize the integration of daylighting strategies and the optimization of natural light penetration[7].

Despite the clear benefits of transparent roofs and walls, traditional glass materials face inherent limitations. Glare, caused by the direct transmission or reflection of sunlight, can lead to discomfort, eye strain, and reduced visual clarity[8,9]. This can significantly impact productivity and overall well-being, especially for individuals who work in environments with excessive sunlight or bright lighting. Privacy concerns arise due to the transparency of traditional glass, particularly in buildings with sensitive functions such as hospitals. Additionally, the issue of excessive heat accumulation inside the building during the summer months necessitates effective heat control mechanisms to ensure a thermally comfortable indoor environment, especially for buildings located in hot, arid countries with many clear sky days. Buildings with transparent roofs and walls consume more electricity for the air conditioning systems compared to normal buildings.

[1]Institute of Microstructure Technology, Karlsruhe Institute of Technology, Hermann-von-Helmholtz-Platz 1, 76344 Eggenstein-Leopoldshafen, Germany.
[2]Light Technology Institute, Karlsruhe Institute of Technology, Engesserstrasse 1, 376131 Karlsruhe, Germany. ✉e-mail: gan.huang@kit.edu; bryce.richards@kit.edu

To effectively minimize glare and privacy concerns, one solution is the use of light-diffusing films (LDF) to convert the direct beam of sunlight into diffuse radiation. LDFs are commonly made by suspending polymer films with transparent micro- or nanoparticles[10–12]. For example, a combination is polymethyl methacrylate suspended with silica nanoparticles[10,11]. By incorporating a 20% volume of silica nanoparticles, the resulting diffuse transmittance can range from 20% to 40%[10]. LDFs typically have a blurry appearance, which helps protect privacy. Porous films and surface-structured films have also proven to be effective LDF materials[13,14]. The diffuse light can also be collected by luminescent materials and transmitted towards photovoltaic cells for electricity generation[15] and fostering heightened photosynthesis rate[16]. Recent advancements in surface engineering have led to the development of self-cleaning coatings for transparent roofs and walls. Researchers have explored the use of superhydrophobic coatings with micro- and nanostructured surfaces, which repel water and prevent the accumulation of dirt and pollutants[17–19].

In the context of building space cooling in summer, passive radiative cooling technologies have rapidly developed and gained significant attention[20–25]. The cold universe (thermodynamic temperature ~3 K) is a heat sink, which can provide sufficient coldness. Radiative cooling devices poorly absorb solar energy (wavelength $\lambda = $ ~0.3–2.5 μm) to avoid solar heating but are able to radiate heat through the earth's atmosphere's long-wave infrared transmission window ($\lambda = $ ~8–13 μm) to the cold universe. A radiative cooling material is most effective if its surface has a low solar absorptivity ($\lambda = $ ~0.3–2.5 μm) to minimize solar heating, and a high mid-infrared thermal emittance ($\lambda = $ ~8–13 μm) to maximize thermal emission to the universe. Micro-photonic structures (such as micro-cubes, -pyramids, -grooves) have been proven to be able to improve the mid-infrared thermal emittance for radiative cooling[26–29]. While most radiative cooling materials are opaque[30–34], recent studies have focused on developing translucent radiative cooling materials, which allow the sunlight to pass through and also can provide cooling capacity ranging from ~50 to ~100 W/m$^2$ [2,35–41].

More recently, advancements have been made in the development of emerging multi-functional materials to solve the aforementioned challenges faced by traditional transparent roofs and walls. A translucent nanocellulose metamaterial film, comprising cellulose nanofibrils and $SiO_2$ microparticles, is capable of integrating radiative cooling and light diffusion[37]. This material exhibited a total transmittance of approximately 90%, diffused transmittance of around 70%, and an emissivity of approximately 0.93 in the 8–13 μm range. While several studies have focused on materials integrating radiative cooling and self-cleaning functionalities[42–46], most of them rely on opaque nano-/micro-porous structures, limiting their transparency to visible light[42–45]. An alternative transparent metamaterial composed of ZnO nanorods and $SiO_2/TiO_2$ nanoparticles, demonstrates a high contact angle of 155° for self-cleaning, albeit with a lower transmittance of 73%[46]. Despite these recent advancements, the existing transparent materials still struggle to effectively address simultaneous challenges of glare, privacy concerns, and overheating issues.

Here, our objective is to develop a multi-functional metamaterial that efficiently and simultaneously addresses the challenges. To achieve this, we investigate a polymeric metamaterial with a surface structure composed of micro-pyramids measuring approximately 10 μm in width. The micro-pyramid structures exhibit sunlight diffusing and minimize reflection losses through multiple reflections, resulting in a transparency of 95% for visible light, and an overall diffuse transmittance of 73%. Furthermore, the micro-pyramid structures lead to an emissivity of 0.98, closely resembling that of an ideal mid-infrared (MIR) black body and enabling passive radiative cooling. Leveraging the micro-pyramids' resemblance to the micro-cones found on lotus leaves, our metamaterial also possesses superior superhydrophobic properties, facilitating self-cleaning capabilities. Importantly, the

metamaterial is constructed from commercially available and environmentally safe polymer materials[47,48], ensuring its potential for large-scale manufacturing while remaining competitive both economically and environmentally with existing transparent roof and wall materials.

## Results

### Polymer-based micro-photonic multi-functional metamaterial (PMMM) concepts

We have developed a polymer-based micro-photonic multi-functional metamaterial (PMMM) featuring micron-scale square-based pyramid surface structures, as shown in Fig. 1a. This design integrates several functionalities, including light diffusing, self-cleaning, and radiative cooling, while maintaining a high level of transparency. The key aspect of the PMMM film lies in its ability to efficiently radiate heat through the long-wave infrared transmission window of the Earth's atmosphere (wavelength $\lambda = $ ~8–13 μm), effectively releasing heat into the cold expanse of the universe. This feature allows for passive radiative cooling without any electricity consumption.

To achieve these properties, we carefully selected polydimethylsiloxane as the base material for fabricating the PMMM. This choice was driven by its transparency, cost-effectiveness, environmental safety, stability[49,50], and its high emissivity in the crucial wavelength range of 8–13 μm[51]. Compared to the traditional soda-lime window glass, which appears transparent, the PMMM sample (coated on a glass substrate) exhibits a blurry appearance, as shown in Fig. 1b. This characteristic arises from the PMMM film's light-diffusing capabilities, effectively concealing the background. Consequently, when utilized for constructing roofs or walls, PMMM offers a means to safeguard users' privacy. The PMMM features micro-pyramid units measuring approximately $w = 10$ μm in width, set at an inclined angle of 54.7°, as shown in Fig. 1c. The PMMM design incorporates a density of 1 million such micro-pyramids per square centimeter.

In conventional glass roofs and walls, incident sunlight is split into direct transmitted light and direct reflected light, as shown in Fig. 1d. Unfortunately, this separation can lead to issues such as uncomfortable glare, creating an unsuitable indoor lighting environment. Moreover, direct reflected light can also lead to glare problems to the surroundings. Transparent glass roofs and walls exacerbate the greenhouse effect, elevating indoor temperatures and necessitating higher energy consumption by air conditioning systems compared to opaque roofs. In contrast, the PMMM roof offers a solution by effectively diffusing both transmitted and reflected light, addressing glare problems, as shown in Fig. 1e. The diffuse light also enhances photosynthesis rates in some plants. With its superhydrophobic properties, the PMMM can be cleaned by rain or dew water, reducing the need for frequent maintenance compared to traditional glass roofs. Furthermore, the PMMM's radiative cooling effect provides additional cooling for buildings with translucent roofs, effectively lowering room temperatures and consequently saving electricity consumption by air conditioning systems.

### Light management and radiative cooling performance of PMMM

The process of fabricating the PMMM film mainly includes photolithography, etching, solution casting, and pealing, as shown in Fig. 2a. We begin by selecting a (100)-oriented silicon wafer coated with a 1-μm-thick $SiO_2$ layer as the base to create the casting mold. The pattern of inverted micro-pyramids on the silicon wafer is fabricated using photolithography and wet etching techniques. Subsequently, the polydimethylsiloxane solution, mixed with a curing agent, is cast onto the silicon mold through blade coating. After heat curing, the flexible and resilient PMMM film is carefully peeled off from the mold. A detailed fabrication process is available in the Methods section and Supplementary Fig. 1. For applications requiring added strength, the PMMM film can be flattened and adhered to a glass substrate. Polydimethylsiloxane naturally exhibits excellent adhesion to glass

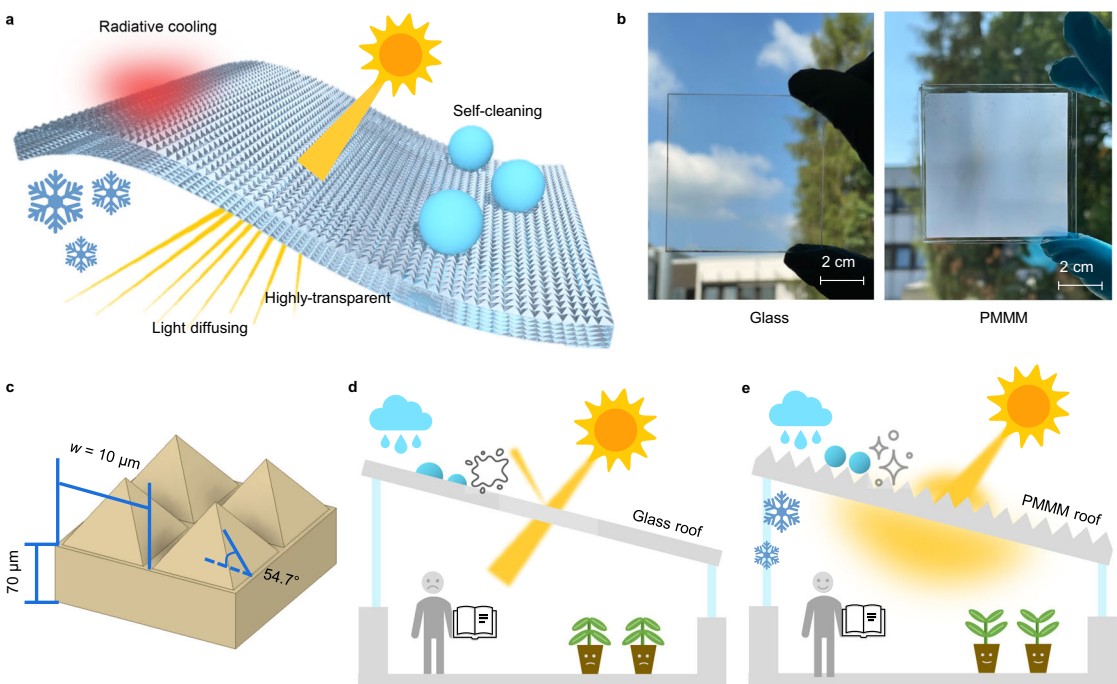

**Fig. 1 | Schematic illustration of the polymer-based micro-photonic multi-function metamaterials (PMMM) for integrated light, thermal and wettability management. a** Concept of PMMM integrated the functions of light diffusing, anti-reflection, self-cleaning and radiative cooling. **b** Photographs of a soda-lime glass and a PMMM film on a soda-lime glass substrate. **c** The structure and size of micro-pyramid structures. **d** A building with a traditional glass roof. **e** A building with the PMMM roof. The PMMM roof has less reflection loss, and significant light diffusing, resulting in a more comfortable light environment. The plants in the building will also grow faster and healthier, such as tomato plants. The stronger radiative cooling performance of the PMMM roof can provide effective electricity-free radiative cooling.

(as detailed in Supplementary Fig. 2). Moreover, scaling up the size of the PMMM film can be conducted by increasing the lithography area, as demonstrated in Supplementary Fig. 3. In our experiments, we successfully upscaled the PMMM film size from 3.2 cm² to 42.2 cm², as shown in Fig. 2b. To exemplify its scalability, a single piece of PMMM film can reach a maximum size of approximately 450 cm² (approximately 21.2 cm × 21.2 cm) by employing a 12-inch silicon wafer to fabricate the mold. By assembling multiple PMMM cells together, similar to creating large photovoltaic panels composed of numerous photovoltaic cells, the PMMM film can be upscaled to the square meter scale. The PMMM film can be further upscaled by using a matrix of molds or hot embossing technology, as detailed in Supplementary Fig. 4.

The optical properties of traditional 1-mm-thick soda-lime glass and PMMM in the wavelength ranges of 0.3–2.5 µm (spectral range of terrestrial sunlight) and >4 µm (thermal radiative emission spectrum range) are compared in Fig. 2c, d. Illumination and light collection geometries of the spectrophotometer are shown in Supplementary Fig. 5. For the glass, the global transmittance in the sunlight spectrum range has a spectrally weighted average value of 91% (The methods of calculating the spectrally weighted average values is detailed in Method section). Additionally, the average diffused transmittance and reflectance of the glass are only both about 1%, meaning that the majority of sunlight is either directly transmitted or reflected. Notably, the glass exhibits a significant emissivity valley in the 8–13 µm range, resulting in an emissivity value of 0.87. In contrast, the PMMM film significantly improves the optical performance of traditional glass, as shown in Fig. 2d. The average global transmittance of the PMMM in the sunlight spectrum range is 95%, surpassing that of glass (91%). This is because the incident rays reflected at the micro-pyramid's surfaces can be redirected to the PMMM[52]. The light rays reflecting off the inclined surfaces of the micro-pyramids are redirected towards adjacent micro-pyramids. This redirection causes the reflected rays to be transmitted

deeper into the PMMM, thereby enhancing its transmittance. Furthermore, the average transmittance in the visible range reaches 95%, while $\tau_{dif}$ is high at 73%, implying that 90% of the transmitted light is diffused light. The incident light rays that strike the flat areas between the micro-pyramids travel straight through the PMMM without contributing to diffusion. Therefore, $\tau_{dif}$ is closely linked to the proportion of incident light that interacts with the micro-pyramids relative to the total incident light. Notably, the transmitted light is not randomly diffused like a normal light diffuser. The transmitted light diffuses within ±27° (Fig. 2e, f), which depends on the surface microstructure of PMMM. The refraction angle broadens from 18° to 41° when the micro-pyramid's base angles range from 40° to 70°, as shown in Supplementary Fig. 6. The PMMM film also demonstrates an average emissivity in the range of 8–13 µm, reaching as high as 0.98, which is nearly equivalent to the ideal black body. As a result, it efficiently emits heat radiation into space through the Earth's atmosphere transmittance window. The key performance indicators of glass and PMMM are summarized in Fig. 2g (the average global transmittance $\tau_{g\_ave}$ and diffuse transmittance $\tau_{dif\_ave}$ are the spectrally weighted values in the range of 0.3–2.5 µm for the air-mass 1.5 (AM1.5) standard solar spectrum; the average global visible transmittance $\tau_{vis\_ave}$ is the spectrally-weighted value in the visible range of 0.38–0.7 µm; the emissivity $\varepsilon_{ave}$ is the average value in the range of 8–13 µm), showcasing PMMM's superior advantages across all these metrics. The methods of calculating the values of $\tau_{g\_ave}$, $\tau_{vis\_ave}$, $\tau_{dif\_ave}$ and $\varepsilon_{ave}$ are detailed in the Method section. The relative uncertainties of the transmittance and emissivity estimation are 0.4 rel.% and 5.5 rel.% accordingly. Both random (based on a measurement noise and ten times repeated measurements, shown in Supplementary Fig. 7) and systematic uncertainties were taken into account.

Here, we provide an insight into the optical properties of PMMM in different wavelength ranges: the MIR range for radiative cooling and the visible range for sunlight distribution. Firstly, the absorption

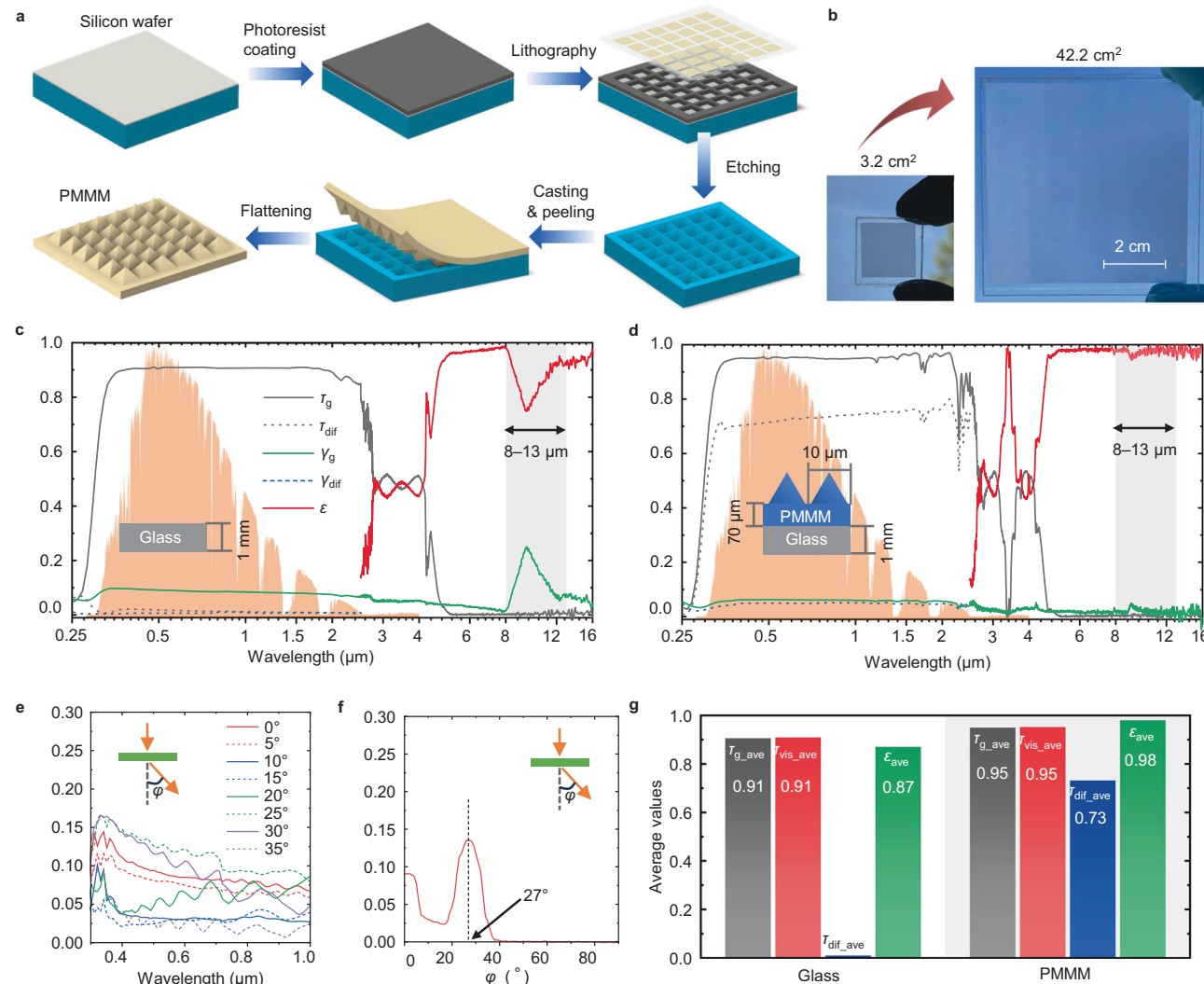

**Fig. 2 | Fabrication of polymer-based micro-photonic multi-function metamaterials (PMMM) and measured optical properties. a** Fabrication process of PMMM. A silicon mold with inverted micro-pyramids is fabricated by photolithography and etching. PMMM film is fabricated by casting polydimethylsiloxane on the silicon mold via blade coating. **b** Photographs of small-scale (3.2 cm²) and larger-scale (42.2 cm²) PMMM films. **c** Optical properties (global transmittance $\tau_g$, diffuse transmittance $\tau_{dif}$, global reflectance $\gamma_g$, diffuse reflectance $\gamma_{dif}$, emissivity $\varepsilon$) of normal soda-lime glass. The background of the figure is the normalized AM1.5 G solar spectrum. The average global transmittance of the glass is 92%, with diffused transmittance less than 1%. A significant emissivity valley shows in the 8–13 μm range. **d** Optical properties of PMMM film on a 1-mm-thick soda-lime glass

substrate. The average global transmittance of the PMMM sample is 95%, with diffused transmittance as high as 73%. The emissivity (8–13 μm) is ~0.98, which is close to an ideal black body. **e** Angular transmittance of PMMM for angles of 0 to 35°. The transmittance for angle >40° is close to zero, thus is not visible in this figure. **f** Measured polar transmittance distribution of the PMMM sample for 500 nm wavelength. The transmitted light diffuses within ±27°. **g** Comparison of key performance indicators ($\tau_{dif}$ is the global transmittance in the visible spectrum) of the glass and PMMM sample (measured results). The PMMM sample has better performance in direct transmittance, diffused transmittance, and emissivity. Source data are provided as a Source Data file.

properties of the PMMM in the MIR range are described. The absorption properties of our PMMM in the MIR range can be described based on previous studies[53,54]. In the MIR range, the absorption properties of a material are a key parameter to achieve high cooling performance. Polydimethylsiloxane has been recognized as a representative material for radiative cooling, since it intrinsically possesses strong absorption in the atmospheric transparency window (8–13 μm), which is attributed to the fact that the molecular vibrations of Si−CH₃ and Si−O−Si located inside the atmospheric transparency window[53]. Thus, polydimethylsiloxane has been utilised for radiative cooling, especially to develop a scalable radiative cooling system[54]. A flat film made of polydimethylsiloxane can be a good emitter for radiative cooling; however, its absorption capability is limited by reflection occurring at an air-polydimethylsiloxane interface (Supplementary Fig. 8). This reflection originates from the fact that the refractive index of

polydimethylsiloxane dynamically changes at around the resonance wavelengths of the molecular vibrations, which causes large refractive index contrast at the interface. Therefore, the flat polydimethylsiloxane film cannot be an ideal black body in the wavelength range of 8–13 μm. In previous works, micro-pyramid structures have been utilized to allow the system's refractive index to gradually change from the refractive index of air to the refractive index of polydimethylsiloxane[26,27]. This interpretation based on the refractive index gradient can be applied for subwavelength micro-pyramids. However, the micro-pyramid size of the PMMM is comparable to the wavelengths in the MIR range. In this case, resonance modes of the micro-pyramid must be considered to understand the PMMM's absorption mechanism, which can be done by utilizing Cartesian multipole decomposition (CMD)[55]. According to our resonance mode analysis based on CMD (Supplementary Fig. 9), it is found that the

micro-pyramid with $w = 10\ \mu m$ supports a strong electric dipole resonance, and weak magnetic dipole, electric quadrupole, and magnetic quadrupole resonances. Since the micro-pyramid is placed on the polydimethylsiloxane film, those resonance modes can couple to the polydimethylsiloxane film, which can cause strong forward scattering into the polydimethylsiloxane film[56,57]. Consequently, the reflection at the air-polydimethylsiloxane interface diminishes, more energy of the incident wave can be transmitted into the polydimethylsiloxane film, and the PMMM' absorption in MIR range is thus enhanced based on Beer's law.

In the visible range, the PMMM acts as a filter to distribute incoming sunlight inside buildings. Its light distribution properties can be described by utilizing the geometrical optics approximation since the micro-pyramid units of the PMMM (~10 μm) are sufficiently larger

than the visible wavelengths. Figure 3a shows the ray-tracing simulation results of the PMMM for different incident angles ($\theta_{in}$). The orientation of the light source and the micro-pyramids can be found in the inset of Fig. 3a. The color of the rays shows the relative intensity of the rays (the incident rays' intensity is taken as 100%). When $\theta_{in} = 0°$, the transmitted rays (below the bottom glass substrate) are distributed in five directions. One direction is attributed to the fact that the incident rays strike the flat surface between the micro-pyramids and pass through the PMMM straight. The other four directions are leaned from the normal of the PMMM's bottom surface because of refractions occurring at the surfaces of the micro-pyramid structures. Since the transmitted rays propagating in those four directions are connected to $\tau_{dif}$, they can be regarded as diffused rays. The leaning angle of the diffused rays, namely the diffusion angle $\varphi$ (Fig. 2f), is predominantly

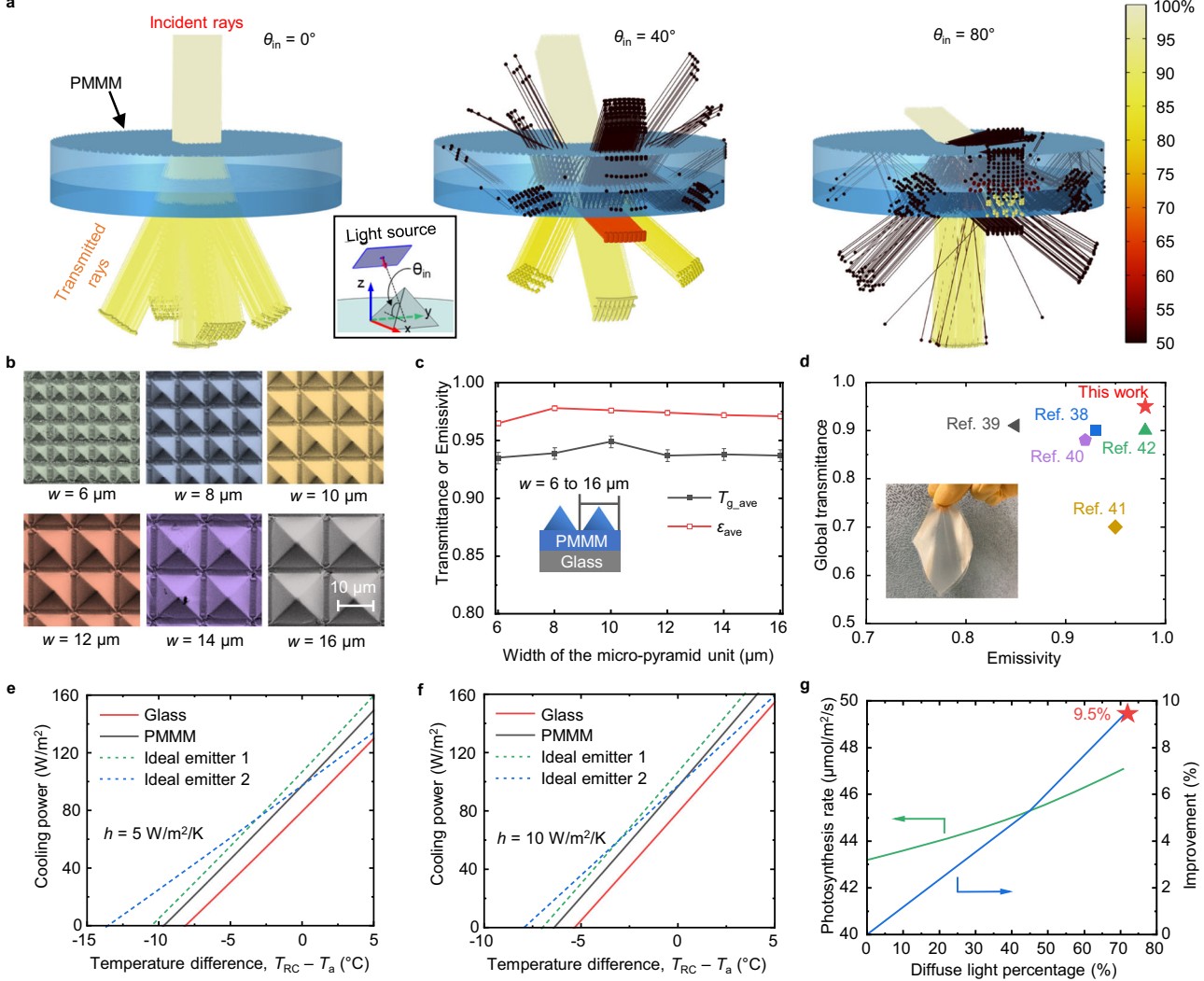

**Fig. 3 | Polymer-based micro-photonic multi-function metamaterials (PMMM) working principle, radiative cooling performance and light diffusion.** **a** Simulated ray propagation maps of the PMMM for $\theta_{in} = 0°$, 40°, and 80° in the visible range. Most of the incident light (>80%) diffuses in the range of ±30° when $\theta_{in} = 0°$. The micro-pyramid structures can successfully redirect the incident sunlight to the down direction when the incident angle is as high as $\theta_{in} = 80°$. **b** SEM images of PMMM samples with different sizes of the micro-pyramids. There are ~1 to 1.5 μm spaces between the micro-pyramids. **c** Measured global transmittance ($\tau_{g\_ave}$) and emissivity ($\varepsilon_{ave}$) of for PMMM with different pyramid sizes. **d** Comparison of the global transmittance (0.3–2.5 μm) and emissivity (8–13 μm) of the PMMM (10-μm pyramid with glass substrate) in this study and transparent/translucent radiative cooling materials in the literature. The PMMM in this study

exhibits both higher transmittance and also higher emissivity than other emerging transparent/translucent radiative cooing materials. The inserted photograph shows the flexible PMMM film. **e** Simulated radiative cooling power of PMMM (10-μm-pyramid with glass substrate) and glass when heat convection coefficient with the ambient is $h = 5\ W/m^2/K$ ($T_a = 25°C$). **f** Simulated radiative cooling power of PMMM (10-μm-pyramid with glass substrate) and glass when $h = 10\ W/m^2/K$ ($T_a = 25°C$). **g** The Photosynthesis rate of tomato plants as a function of diffused light percentage, and the estimated improvement of photosynthesis rate by the PMMM. The PMMM is calculated and estimated to be able to improve the photosynthesis rate by ~9% compared to normal glass. The data of the photosynthesis rate as a function of the diffuse light percentage is referred to experimental results in ref. 61. Source data are provided as a Source Data file.

determined by the base angle (54.7°) of the micro-pyramid structure (Supplementary Fig. 10). The simulation result and a theoretical analysis based on Snell's law give $\varphi = 27°$ (Supplementary Fig. 6 and Supplementary Fig. 10). This value agrees well with the experimental result shown in Fig. 2f. With increasing $\theta_{in}$ from 0° to 40°, the transmitted rays, which can be found below the bottom glass, exit the PMMA at a higher angle. When $\theta_{in}$ exceeds a certain angle, the incident rays can be redirected into the down direction (-z direction) by total internal reflections occurring in the pyramid structures (Supplementary Fig. 10). The ray-tracing simulation model is detailed in Supplementary Fig. 11. Detailed ray propagation maps taken from different directions (0°, 20°, 40°, 60°, 80°) can be found in Supplementary Fig. 12. We also simulated the global and diffused transmittances of the PMMM at $\theta_{in} = 0°$, as shown in Supplementary Fig. 13. The simulated global and diffused transmittances are 93.3% and 72.2%, which are close to the experimental results of $95 \pm 0.4\%$ and $73 \pm 0.3\%$. Diffused rays are generated by the refraction and reflection of light at the surfaces of the micro-pyramids.

We fabricated PMMM samples with different geometric parameters ($w = 6-16$ μm). Figure 3b presents Scanning Electron Microscope (SEM) images of the PMMM samples with various pyramid sizes, where the micro-pyramid size was determined by the photomask design used in the lithography process. It is important to note the presence of small, unavoidable gaps (~1–1.5 μm) between these pyramids. Figure 3c illustrates the influence of the micro-pyramid size on both the emissivity and transmittance characteristics of the PMMM. The variations in both transmittance and emissivity across pyramid sizes ranging from 6 to 16 μm are relatively minor, approximately 0.01. This suggests that PMMM's performance exhibits a strong tolerance to variations in pyramid size. As a result, the need for precise control over pyramid dimensions during fabrication is reduced, underscoring the material's robustness and adaptability in manufacturing processes. To investigate this effect, we used the wave-optics module in COMSOL Multiphysics to study the electric field distribution around the micro-pyramid at its resonance wavelength[57], as detailed in Supplementary Fig. 9. We discovered that the incoming wave interacts with the resonance modes of the micro-pyramid structure, leading to its confinement and efficient transfer to the polydimethylsiloxane film. This happens because the micro-pyramid's base directly contacts the polydimethylsiloxane film, which has a higher refractive index than air, facilitating the wave's coupling to the film[56,57]. Smaller micro-pyramids enhance transmission effectively through this coupling mechanism. Larger micro-pyramids primarily use their forward scattering to strengthen transmission due to their complex resonance states[58]. Therefore, the material can achieve high transmission across its interface for any micro-pyramid size in this study, ensuring size-independent high emissivity in the MIR range. Detailed transmittance and emissivity curves for different PMMM samples are shown in Supplementary Fig. 14. The global transmittance and emissivity of the translucent PMMM (10-μm pyramid with glass substrate) in this study are compared with emerging transparent/translucent radiative cooling materials in the literature[38–42], as shown in Fig. 3d. The inserted image in Fig. 3d shows the flexibility of the PMMM film without a glass substrate. The PMMM sample shows both higher transmittance and emissivity than existing transparent/translucent radiative cooling materials.

The radiative cooling power of the PMMM film is simulated using the mathematical model by Raman et al.[21,59]. The radiative cooling powers of the PMMM and glass samples are plotted against the temperature difference between the radiative cooling sample and the ambient (i.e., $T_{RC} - T_a$) in Fig. 3e, assuming a heat convection coefficient of $h = 5$ W/m²/K (representing a no-wind situation). The radiative cooling power of the PMMM and glass exhibits a linear increase as the working temperature of the radiative cooling sample rises. At ambient temperature, the PMMM film achieves a radiative cooling power of

97 W/m², surpassing the glass radiative cooling power by approximately 18 W/m². Consequently, the PMMM film is capable of cooling to around 10 °C below the ambient temperature through radiative cooling. For comparison, Fig. 3e also includes the radiative cooling performance curves of two types of ideal emitters: Ideal emitter 1 ($\varepsilon = 1$ for $\lambda > 4$ μm and $\varepsilon = 0$ for all other wavelengths) and Ideal emitter 2 ($\varepsilon = 1$ for $\lambda = 8-13$ μm and $\varepsilon = 0$ for all other wavelengths). The radiative cooling powers of Ideal emitter 1 and Ideal emitter 2 at ambient temperature are 107 W/m² and 97 W/m², respectively, higher than that of the PMMM film by 0–10 W/m². Both Ideal Emitter 1 and Ideal Emitter 2 can achieve minimum temperatures of approximately 11 °C and 13 °C below the ambient temperature, respectively. The radiative cooling powers of the PMMM, glass, and the ideal emitters are compared under an ambient heat convection coefficient of 10 W/m²/K in Fig. 3f (representing light wind situations with wind speeds of 1–3 m/s[60]). The PMMM film demonstrates radiative cooling performance closer to that of the ideal emitters when there is light wind compared to no-wind situations. The atmospheric transmittance and model validation are detailed in Supplementary Fig. 15. Thermal radiation exchange within the 3–50 μm range is considered in our numerical model, as shown in Supplementary Fig. 16. The detailed modeling method is detailed in "Method" section.

Additionally, apart from the radiative cooling performance, the PMMM film's light diffusion effect is estimated to enhance the photosynthesis ratio by ~9%, as shown in Fig. 3g. This estimation is based on experimental data from ref. 61, which establishes a relationship between the photosynthesis ratio and the percentage of diffused light (further details provided in Supplementary Fig. 17). Given the PMMM's diffused transmittance of 73%, the average photosynthesis rate for tomato plants is projected to be approximately >50 μmol/m²/s, representing a notable ~9% increase compared to photosynthesis rates without light diffusion.

Diffusion enables a more uniform light distribution, allowing plants to utilize light more effectively for growth. Beyond agriculture, the advantages of diffused light extend to several domains, enhancing both functional and esthetic aspects of various environments. In architectural applications, for instance, diffused light mitigates glare and provides balanced indoor illumination, which contributes to visual comfort and reduces eye strain. This even light distribution is beneficial in workplaces, educational settings, and healthcare facilities, where it supports human performance and well-being. Moreover, diffused light can enhance privacy by softening and scattering direct views through transparent media, an aspect particularly relevant to the PMMM film. The high diffusion transmittance of the PMMM film not only serves esthetic purposes but also promotes privacy and creates a sense of openness without compromising on light quality or energy efficiency. In retail and exhibition spaces, such as art galleries and museums, diffused light plays a pivotal role in presenting products and artworks in a way that minimizes reflections and hotspots, ensuring that each item is viewed under optimal lighting conditions.

The outdoor radiative cooling performance of PMMM samples was evaluated at the Solar Park within the Karlsruhe Institute of Technology campus on 7 December 2023, under clear sky conditions, as depicted in Fig. 4a. The diagram of the testing platform setup is shown in Fig. 4b. All details of the outdoor testing platform setup (dimensions and materials etc.) can be found in the Method section. A cooling test chamber was utilised to test the temperature reduction of the PMMM sample (42.2 cm²) with radiative cooling. This chamber was shielded by a low-density polyethylene (LDPE) film to serve as a windshield. The chamber's interior was lined with a highly reflective film, while its exterior was insulated thermally. To further assess the cooling efficacy, two boxes were constructed: one with a PMMM roof and the other with a normal glass roof. These boxes were designed to absorb sunlight on their inner walls, allowing for a comparative analysis of the PMMM and glass roofs' cooling performances. The

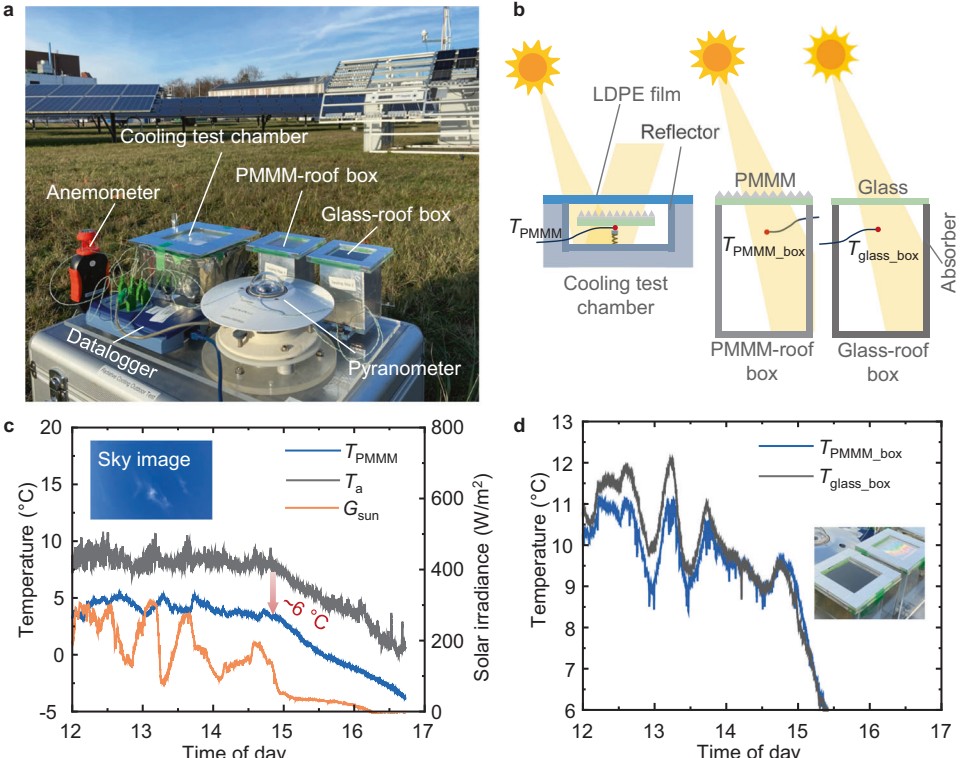

**Fig. 4 | Outdoor testing of radiative cooling performance. a** Photograph of the testing platform within the Solar Park of Karlsruhe Institute of Technology. The cooling test chamber was used to test the minimum temperature of the polymer-based micro-photonic multi-function metamaterials (PMMM) sample with radiative cooling. The PMMM-roof box and glass-roof box were used to test the temperature of the experimental enclosures with PMMM and normal glass as the roofs. **b** Schematics of the cooling test chamber, PMMM-roof box, and glass-roof box. The

cooling test chamber was covered with a low-density polyethylene (LDPE) film as the windshield. Detailed dimensions are provided in the Method section. **c** PMMM sample temperature ($T_{PMMM}$), ambient temperature ($T_a$) and global solar irradiance ($G_{sun}$). The inserted image shows the sky at 13:00. **d** Temperature of the PMMM-roof ($T_{PMMM\_box}$) and glass-roof boxes ($T_{glass\_box}$). The inserted image shows the enlarged view of the two boxes. Source data are provided as a Source Data file.

temperatures of PMMM, the boxes, and the ambient were monitored using thermocouples.

The day of testing featured a clear sky as depicted in Fig. 4c. The average humidity was approximately 63% between 12:00 and 15:00. Throughout the test, the temperature of the PMMM was consistently lower than the ambient temperature. At 14:48, the PMMM temperature was ~6 °C lower than the ambient temperature, demonstrating significant cooling in the high-humidity environment of Karlsruhe due to PMMM's strong emissivity. Remarkably, at 16:30, post-sunset, the PMMM temperature dropped to −2.8 °C, well below the freezing point, while the ambient temperature was 0.5 °C.

Comparatively, the PMMM-roof box maintained a lower temperature than the glass-roof box until 14:40, attributed to PMMM's superior thermal emissivity, as shown in Fig. 4d. At 13:13, the temperature in the PMMM-roof box was 1.7 °C lower than that in the glass-roof box, highlighting the more effective radiative cooling power of the PMMM roof. Interestingly, after 14:40, the temperature in the PMMM-roof box exceeded that of the glass-roof box. This change was due to the micro-pyramid structure's ability to direct sunlight into the box at lower sun elevation angles in the late afternoon (the elevation angle of the sun was ~10° at 14:40), as illustrated by the ray-tracing results in Fig. 3a. The inserted image offers a view of both the PMMM-roof and glass-roof boxes, showing the PMMM's distinctively blurred and colorful appearance at certain angles.

## Self-cleaning performance of PMMM

The natural lotus leaf's self-cleaning performance stems from its superhydrophobic surface, featuring micro-cone structures of approximately 5–10 μm, similar to the micro-pyramids on the PMMM

surface. The contact angles ($\theta$) of various surfaces are measured and compared in Fig. 5a. The glass exhibits hydrophilic behavior with $\theta = 26°$, while planar polydimethylsiloxane is hydrophobic with $\theta = 110°$. The PMMM sample with spaces between adjacent pyramids of $\delta = -1.5$ μm exhibits a higher contact angle of 152° compared to the 128° of the sample with $\delta = -4$ μm has a contact angle of 128°. This behavior aligns with the principles of the *Wenzel* equation, as detailed in Supplementary Fig. 18. The matrix of the micro-pyramids on the PMMM surface is shown in Fig. 5b. The micro-pyramid matrix can "lift" water droplets, leading to an increased contact angle, as shown in Fig. 5c. As detailed in Supplementary Fig. 14, we also examined the contact angles for the samples with pyramid unit size varying from 6 to 16 μm. The variation in contact angle is less than 3° across the range of pyramid sizes, translating to a change of less than 2%.

In the experimental setup, a few real dust particles were collected from a local window in Karlsruhe, Germany. The SEM image in Fig. 5d shows that most of the dust particles are larger than 10 μm, with a majority being silicate and carbonate minerals (Supplementary Fig. 19), which are common constituents of atmospheric dust. Figure 5e demonstrates the active rain self-cleaning mechanism of PMMM, providing efficient cleaning in the rain or manually sprayed water. On hydrophilic glass surfaces, water droplets tend to adhere and flow down slowly, resulting in low cleaning efficiency. In contrast, on the PMMM surface, water droplets swiftly jump and sweep through (Supplementary Movie 1), enabling efficient removal of dust particles from the surface. To verify the active self-cleaning ability, an indoor simulated experiment is conducted, as shown in Fig. 5f. The dust is intentionally deposited on the 45°-inclined PMMM surface. Subsequently, a continuous stream of water droplets is released onto the

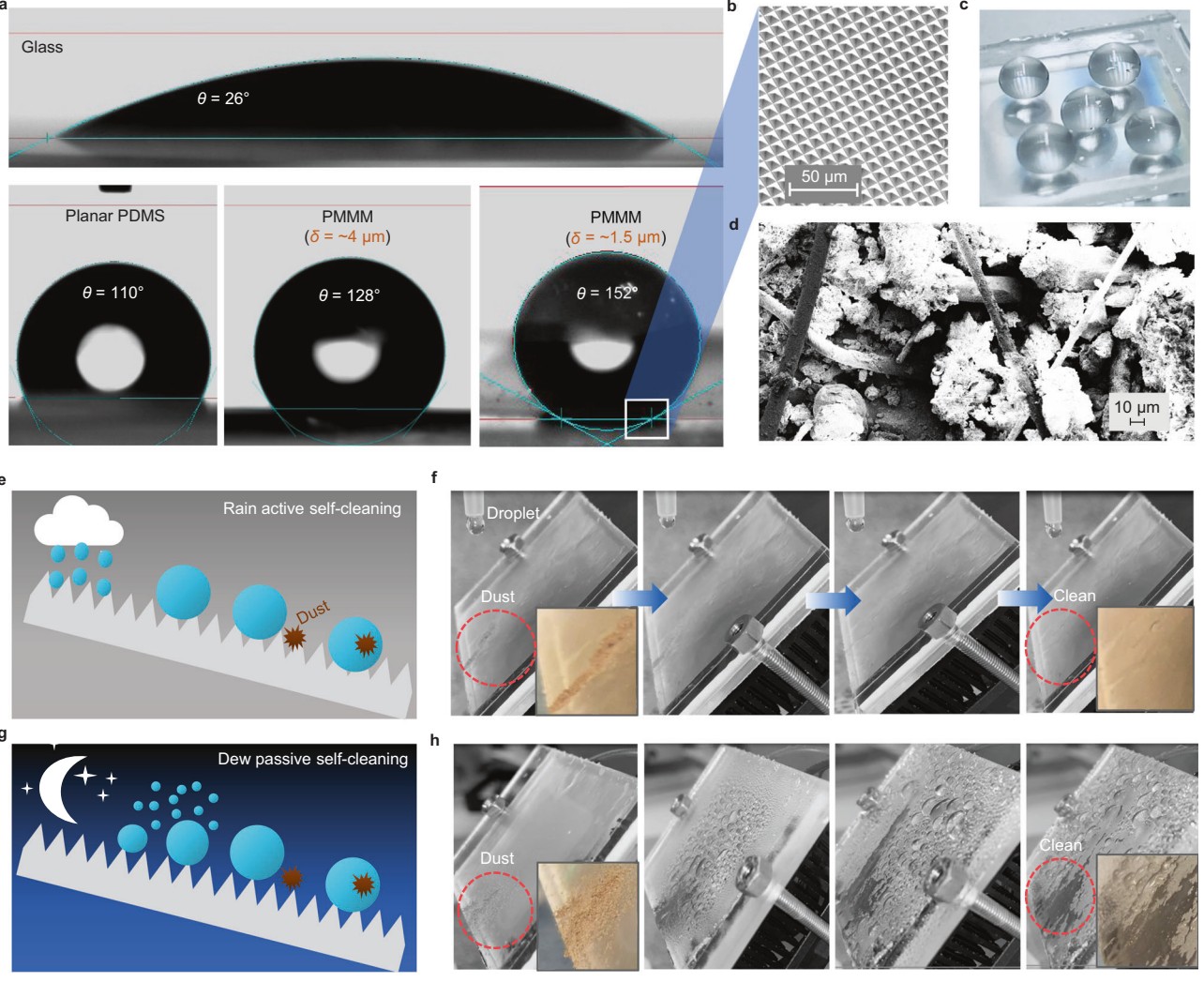

**Fig. 5 | Self-cleaning performance of polymer-based micro-photonic multi-function metamaterials (PMMM).** **a** Contact angles of glass, flat polydimethylsiloxane, PMMM (4-μm-wide space between adjacent micro-pyramids) and PMMM (1.5-μm-wide space between adjacent micro-pyramids). **b** SEM image of the PMMM. **c** Superhydrophobic phenomenon. **d** SEM image of the dust particles in the self-cleaning testing. Most dust particles are larger than 10 μm. **e** Schematic of rain active self-cleaning. **f** Active self-cleaning process (indoor lab simulated experiment). **g** Schematic of dew passive self-cleaning. **h** Passive self-cleaning process (indoor lab simulated experiment).

PMMM surface using a syringe, removing the dust particles and leaving the PMMM surface clean (Supplementary Movie 2). This active self-cleaning feature of the PMMM film presents a significant advantage over traditional surfaces, making it an attractive choice for various applications, including building materials and other outdoor surfaces that require easy maintenance and dust particle removal.

Another interesting aspect of the PMMM film is its passive self-cleaning capability through dew water harvesting, as illustrated in Fig. 5g. A recent modeling study confirms that radiative cooling films can effectively harvest sufficient dew water to clean their surfaces[62]. The strong radiative cooling effect of the PMMM film during nighttime facilitates efficient dew water harvesting. Subsequently, the dew water easily rolls off the superhydrophobic PMMM surface, preventing it from sticking and facilitating the removal of dust particles. To demonstrate and simulate the passive self-cleaning process, an indoor experiment was designed, as shown in Fig. 5h. The PMMM was cooled using a thermoelectric cooler to simulate radiative cooling, achieving a temperature approximately 5° C below the ambient temperature. A humidifier was used to increase the humidity of the ambient air to fasten the condensation progress. As a result, vapor in the air

condensed onto the cooled PMMM surface, forming water droplets. These water droplets then smoothly rolled off through the superhydrophobic PMMM surface, efficiently removing dust particles and leaving the surface clean (Supplementary Movie 3). This indoor experiment showcases the practical application of PMMM's self-cleaning capability, which can be harnessed in real-world scenarios for reduced maintenance and enhanced surface cleanliness. By combining both active and passive self-cleaning processes, the PMMM film remains clean, requiring minimal maintenance when compared to conventional glass surfaces. The ability of the PMMM film to self-clean through both active rain self-cleaning and passive dew water harvesting makes it a practical and low-maintenance option for various applications. It offers significant advantages in reducing the need for frequent cleaning and ensuring long-term performance and durability in real-world environments. We also conducted comparative experiments on a glass surface. As demonstrated in Supplementary Fig. 20a, the glass surface, being hydrophilic, struggles with active cleaning by water droplets. We used the same amount of water droplets to clean the glass surface as we used to clean the PMMM samples. The droplets tend to adhere to the glass, making it difficult to effectively dislodge

dust with the limited amount of water droplets. More water is required to remove the dust further and fully on the glass surface. For the dew cleaning as shown in Supplementary Fig. 20b, the hydrophilic nature of the glass surface leads to the formation of a dew film instead of discrete droplets, lacking the roll-off effect necessary for effective cleaning. As a result, the wetted dust remains adhered to the glass surface, making it more challenging to remove effectively at the end of the dew formation process. In contrast, the hydrophobic PMMM surface easily forms noticeable dew droplets that can roll off, sweeping across the surface and efficiently removing dust.

To provide a more comprehensive assessment of our metamaterial's stability and self-cleaning ability, we conducted a real-world test by exposing samples to the outdoor environment in Karlsruhe, Germany, for durations of one and two months. Specifically, the 1-month sample was exposed from 14 November 2023 to 15 December 2023, while the 2-month sample was left outdoors from 14 November 2023 to 15 January 2024. Post-exposure, we re-evaluated the metamaterial's optical performance. The results, which are elaborated in Supplementary Fig. 21, show a transmittance reduction of 3% after one month. Furthermore, the emissivity displayed a decrease of only 1% during the same period. It's important to note that both transmittance and emissivity underwent only marginal changes with the extension of exposure time to two months. To disentangle the impacts of aging from soiling, the 2-month sample underwent a cleaning process using distilled water. We then reassessed its transmittance and emissivity for a comparative analysis with the pre-cleaned state, as shown in Supplementary Fig. 21c, d. The results revealed negligible changes in both transmittance and emissivity post-cleaning, suggesting that aging, rather than soiling, is the primary factor contributing to the slight reductions observed in these optical properties. This data suggests a relative stability and also self-cleaning ability of the metamaterial under extended outdoor environmental conditions.

## Discussion

We introduced a polymer-based micro-photonic multi-function metamaterial (PMMM) with micro-pyramid surface structures that efficiently and simultaneously address the challenges faced by traditional glass materials. The PMMM film demonstrated a high transparency of 95%, combined with a high diffusion transmittance of 73%. This light diffusing effect effectively minimizes glare while creating a comfortable and visually appealing indoor environment. Furthermore, the PMMM film exhibits an emissivity of 0.98 in the 8–13 μm range, enabling electricity-free radiative cooling for buildings. The PMMM film also shows self-cleaning capabilities. Its superhydrophobic properties are active self-cleaning during rain or water spray. Additionally, its radiative cooling effect allows for passive self-cleaning through dew water harvesting during the nighttime.

The combination of these features makes PMMM a practical solution for transparent roofs and walls, offering improved light management, energy efficiency, and occupant comfort. Moreover, the use of readily-available, affordable, and environmentally-friendly polymer materials ensures the potential for large-scale manufacturing while remaining competitive with existing transparent roof and wall materials. Overall, the development of this multi-functional metamaterial paves the way for sustainable green buildings with enhanced transparency, energy efficiency, and occupant well-being. It contributes to the ongoing efforts towards creating a more sustainable built environment.

## Methods
### PMMM fabrication
**Silicon wafer mold fabrication.** A 4-inch (100)-oriented polished silicon wafer coated with 1-μm-thick silica was selected as the base wafer for the mold fabrication. Negative photoresist (ma-N 1420) was then coated on the silicon wafer by spin coating (3000 rpm for 30 s). The

5-inch chrome photomasks with the designed patterns (fabricated by JD Photo Data, the UK) were used in a photolithographic setup via a mask aligner (EVG620) at 365 nm wavelength. The 1-μm-thick silica coating was selectively etched by reactive ion etching for 6 min with $CHF_3/SF_6$, and then the photoresist is removed by acetone. The silicon wafer was then soaked in 40% KOH solution at 80 °C for 15 min for anisotropic wet etching (etching rate: ~75 μm/h), to reveal the (111) crystal planes that serve as the faces for the inverted micro-pyramids.

**Anti-adhesive coating synthesis.** Twenty microlitres trichloro-(1H,1H,2H,2H-perfluoroctyl)-silane (PFOTS, VWR) was mixed with 10 mL cyclohexane (VWR). The PFOTS solution is then coated on the silicon wafer for 1 h via evaporation coating.

**Polydimethylsiloxane solution casting.** The base silicone elastomer and curing agent (VWR) was mixed with a ratio of 10:1 and stirred for 10 min. The mixture was then put in a vacuum oven for more than 1 h until all the bubbles in the mixture were removed. The polydimethylsiloxane mixture solution was pasted onto the silicon wafer via blade coating. The silica wafer was then heated by a hot plate heater at 90 °C for 15 min. The cured PMMM film was peeled off from the silicon wafer manually.

### Optical characterization
The measurements of the reflection and transmission properties were carried out using UV-Vis-NIR Spectrophotometer (Agilent Cary 7000) across a wide range of wavelengths (300–2500 nm). The instrument was equipped with an integrated sphere with a highly reflective inner surface to collect both diffused and direct light. Direct transmittance at variable angles was measured by the Cary 7000 UMS module. The measurements were recorded and controlled via Cary WinUV software. The emissivity was measured by a Fourier-transform infrared spectroscopy (Bruker Vertex 70) which was equipped with an integrated sphere (A562) with a highly infrared reflective inner surface coated with gold. The below equations are used to calculate the values of $\tau_{g\_ave}$, $\tau_{dif\_ave}$, $\tau_{vis\_ave}$ and $\varepsilon_{ave}$:

$$\tau_{g\_ave} = \int_{0.3\mu m}^{2.5\mu m} \tau_g(\lambda) G_{AM1.5}(\lambda) d\lambda \Big/ \int_{0.3\mu m}^{2.5\mu m} G_{AM1.5}(\lambda) d\lambda \quad (1)$$

$$\tau_{dif\_ave} = \int_{0.3\mu m}^{2.5\mu m} \tau_{dif}(\lambda) G_{AM1.5}(\lambda) d\lambda \Big/ \int_{0.3\mu m}^{2.5\mu m} G_{AM1.5}(\lambda) d\lambda \quad (2)$$

$$\tau_{vis\_ave} = \int_{0.38\mu m}^{0.7\mu m} \tau_g(\lambda) G_{AM1.5}(\lambda) d\lambda \Big/ \int_{0.38\mu m}^{0.7\mu m} G_{AM1.5}(\lambda) d\lambda \quad (3)$$

$$\varepsilon_{ave} = \int_{8\mu m}^{13\mu m} \varepsilon(\lambda) d\lambda / (13 - 8\mu m) \quad (4)$$

where $G_{AM1.5}$ is AM1.5 standard solar spectrum.

### Outdoor testing
The cooling test chamber, constructed from polymethyl methacrylate (PMMA), features a 5-cm-thick layer of polyethylene foam with an aluminum foil coating on its outer wall. This layer is specifically designed to minimize heat transfer with the surrounding environment. The chamber is also equipped with a low-density polyethylene (LDPE) foil as a windshield. Its inner wall is lined with high-reflective reflectors (*Alanod*) to enhance its insulation properties. Both the PMMM-roof box and the glass-roof box are similarly constructed from PMMA and are coated with aluminum reflector foil. Their inner walls are lined with solar absorber foil (*Alanod*). Each sample measured in the experiment has an area of 42.2 cm². The internal dimensions of the cooling test

chamber are 100 mm × 100 mm × 20 mm, while the PMMM-roof box and glass-roof box measure 65 mm × 65 mm × 100 mm each. Temperature measurements were conducted using thermocouples (*Pico*). The ambient temperature was measured by a thermocouple placed in the air near the test platform. The temperature of the PMMM was measured by attaching a thermocouple beneath it. The thermocouple was then subjected to pressure from a spring, as depicted in Fig. 4b. The entire testing platform was positioned horizontally. Solar irradiance was monitored using a pyranometer (*Eppley*), wind speed was gauged with a cup anemometer (*BTMETER*), and relative humidity was recorded using a hygrometer (*ThermoPro*).

### Wettability and self-cleaning characterization
We utilized a contact angle goniometer (with SCA20 Software), which employs three cameras for precise measurements, to measure the contact angles of samples. The sample was placed on a magnetic base. A dosing volume of 1 μl water droplet was dispensed onto the sample. Images were captured to document the contact angle measurements.

### Ray-tracing simulation model
In order to investigate the light distribution properties, we conducted the ray-tracing simulation using COMSOL Multiphysics, which is a commercial software package based on the finite element method (FEM). The simulation model is composed of three parts: the top micro-pyramid structure, middle polydimethylsiloxane film, and bottom soda-lime glass. The model considers that the micro-pyramid units with a width of 10 μm and a gap distance of 1.46 μm are periodically placed on the substrate, whose radius is 300 μm. The thicknesses of the polydimethylsiloxane film and glass substrate are 70 μm and 50 μm, respectively. The model considers that the polydimethylsiloxane micro-pyramids with a width of 10 μm are periodically placed on the polydimethylsiloxane -glass substrate, whose radius is 300 μm. The thicknesses of both films are 50 μm. The pyramids are oriented so that their surfaces face to x- and y-directions. The refractive indices of polydimethylsiloxane and glass are 1.4 and 1.52[63,64], respectively. The size of the light source is 100 μm × 100 μm for normal incidence. The source emits 500 rays. A boundary condition to stop the propagation of the rays outside of the PMMM is applied around the PMMM. A Detailed schematic diagram of the model is presented in Supplementary Fig. 11.

### Radiative cooling power simulation model
In our study, we adopted the energy balance model initially proposed by ref. 21. to simulate radiative cooling power. This model is widely recognized for its effectiveness in calculating the potential radiative cooling power under specific atmospheric conditions, specifically the MODTRAN Standard Atmosphere used in ref. 21. and also this study, characterized by midlatitude, clear skies, and low humidity. Notably, we have integrated a refined atmospheric model recently developed by ref. 59. to enhance the accuracy of our calculations, particularly regarding atmospheric thermal radiation. The updated model distinguishes between two primary atmospheric contributors: Ozone, located higher in the atmosphere, and a combination of $CO_2$ and water vapor, which is distributed throughout the atmosphere. This differentiation is reflected in the formula[59]:

$$I_{atm}(\theta,\lambda) = \varepsilon_{Ozone}(\theta,\lambda) \cdot I_{bb}(T_{Ozone},\lambda) + \varepsilon_{rest}(\theta,\lambda) \cdot I_{bb}(T_{rest},\lambda) \quad (5)$$

where $\varepsilon_{Ozone}$ and $T_{Ozone}$ are the emissivity and temperature of the Ozone layer; $\varepsilon_{rest}$ and $T_{rest}$ are the emissivity and the temperature of the rest of the atmospheric components; $I_{bb}$ is the thermal irradiance of the black body. The detailed raw data utilized for this corrected atmospheric model is accessible in ref. 59.

## Data availability
The data generated in this study are provided in the Supplementary Information and Source Data file, or from the corresponding author upon request. Source data are provided with this paper.

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

## Acknowledgements

This work received the full support from Karlsruhe Institute of Technology (KIT) YIG-Prep-Pro project, and Research Field Energy: Program Materials and Technologies for the Energy Transition (Topic 1 Photovoltaics and Wind Energy, ref. 38.01.05)—as well as the Karlsruhe School of Optics and Photonics (KSOP) is gratefully acknowledged. The authors want to thank the help from Dr. Bharat Sharma from the Institute of Microstructure Technology (IMT) at Karlsruhe Institute of Technology (KIT), who provided valuable advice on the KOH etching parameters. The authors also want to thank Dr. Justine Nyarige (IMT, KIT) who provided help on SEM/EXD measurements.

## Author contributions

G.H. and B.S.R. developed the concept, research methodology, and then directed the project. G.H., A.R.Y., P.P., U.K., D.B. and T.Z. conducted the sample fabrication and optical characterization. K.M. conducted ray-

tracing simulation and CMD. K.T. simulated the cooling power. A.D. and E.A. conducted wettability characterization and self-cleaning experiments. U.K. conducted photolithography and etching. D.B. conducted uncertainty analysis. G.H., K.M., D.B. and B.S.R. contributed to writing and revising the manuscript.

## Funding

## Competing interests

The authors declare no competing interests.

## Additional information

**Publisher's note** Springer Nature remains neutral with regards to jurisdictional claims in published maps and institutional affiliations.

