## [Peer Review File · Nature Communications]

Radiative cooling and indoor light management enabled by a transparent and self-cleaning polymer-based metamaterialREVIEWER COMMENTS

Reviewer #1 (Remarks to the Author):

This work presents a polymer-based micro-photonics multi-functional metamaterial (PMMM), which integrates several functionalities, including light diffusing, self-cleaning, radiative cooling, while maintaining a high level of transparency. And a detailed discussion of special optical properties (especially the light diffusing in the visible range) is given through theoretical simulations and calculations.

1. In line 141-142, "For applications requiring added strength, the PMMM film can be flattened and adhered to a glass substrate." How is the adhesion between the PMMM film and glass, and is it able to maintain a stable bond over time? Is it necessary to add additional auxiliary adhesives in the actual construction, and if so, does it affect the optical properties of the material?
2. Light management and radiative cooling performance of PMMM are mentioned in this work, but only optical properties and theoretical calculations are available to illustrate its potential application in thermal management, lacking real-world demonstrations or related thermal management experiments. It is recommended to add more content for better visualization of experimental results.
3. Figure 2E and F discuss the diffusion angle of transmitted light, which is mainly located within 30 degrees, so is there any connection between this angle and the surface microstructure of PMMM? And is it possible to further increase the diffusion angle of transmitted light, so that the sunlight can be more fully utilized, which will make the indoor lighting environment friendlier?
4. The "Introduction" and "Discussion" sections are too lengthy and could be streamlined to improve readability.
5. The various optical property indexes mentioned in Figure 2G, do they refer to the value at a certain wavelength or the average value in the spectral range? If it involves the calculation of average values (e.g., the average infrared emissivity at 8-13 μm), the relevant formulas and the calculation process should be elaborated in detail in the "Methods" section.

Reviewer #2 (Remarks to the Author):

The authors propose a polymer-based micro-photonics multi-functional metamaterial (PMMM) film of the enhanced radiative cooling characteristics and the anti-dusting feature. Microscale pyramid structures are proposed for the better radiative cooling features. In addition, the self-cleaning properties are shown upon the anti-adhesive coating on the microscale surface structures. The work is interesting, and the manuscript is well-written. However, the following comments should be addressed for further consideration of accepting this manuscript to Nature Communications.

1. In Fig. 3, the ray propagation maps of the PMMA in terms of is interesting like the omnidirectional antireflection properties of moth-eye structures. From that perspective, the structured film seems to exhibit excellent radiative cooling performance upon the simulated incident rays for a day, compared to planar thin film. The authors should provide the estimation on the time-integrated radiative cooling performance upon the simulated incident rays for a day, compared to the planar film to highlight the figure-of-merits of the PMMM film.
2. The radiative cooling power of the proposed PMMM film is just simulated. The experimental results must be accompanied to the simulation results. To support the claims on the structural merits, the experimental results on the radiative cooling are very important, but it is missing.
3. The authors estimated the enhancement of photosynthesis upon the deployment of the PMMA film by ~9%. The enhancement can be attributed to the importance of diffused light for photosynthesis based on the reference. When the authors more elaborate the discussions in terms of the importance of diffused light in some specific applications, it would be helpful to the readers to understand the importance of this study.
4. For anti-dusting characteristics, the information, including size and key element distributions of dusts, on the tested dust particles should be provided, together with the discussions on the mechanism of anti-dusting, for the better understanding and the further optimization of self-

cleaning and anti-dusting properties.

5. Finally, wafer-scale fabrication of the PMMM film was successfully achieved, but for the real application on the roof, much larger-scale fabrication is required. Further developments should be discussed in detailed manner for practical aspects of the PMMM film.

Reviewer #3 (Remarks to the Author):

Summary

In their manuscript "Radiative cooling, light management and self-cleaning based on polymer metamaterials," the authors propose a polymer-based micro-photonics multi-functional metamaterial (PMMM) as a novel building material for transparent roofs and walls. Their PMMM exhibits high, diffuse transmittance to visible wavelengths and high infrared emittance in the atmospheric transmission window. Further, the microstructure of the PMMM results in a superhydrophobic surface, enhancing its self-cleaning properties. All-in-all, the advantages of such a material sound promising as an alternative building material.

While I believe the work would be appropriate and interesting to the community of Nature Communications readers, the manuscript itself needs revision before it is acceptable for publication. To that end, I have included below 14 comments which should be addressed before this work is ready for release. I am optimistic that the authors can satisfactorily address my concerns and communicate their excellent science but, until that time, I cannot recommend publication.

Comments

Comment 1

One of the most important functions of more traditional, opaque passive radiative coolers is to reject as much incoming solar radiation as possible. This is due to the sheer magnitude of the incoming solar irradiance compared to how much heat a blackbody can emit at terrestrial temperatures. In the case of this metamaterial, transparency is valued since the sunlight can be used to offset lighting needs. Relative to glass, the metamaterial transmits more solar radiation into the structure. That means more solar energy will eventually be thermalized in the structure, increasing the cooling needs of the building. The authors mention the problem of excess heat accumulation starting on line 39, but then promote a technology that, at first glance, would seem to make the problem worse. I would like to see a discussion by the authors addressing this. Is there an optimal transmittance to balance cooling and lighting energy usage, or is it always better to transmit as much or as little as possible?

Comment 2

On line 247, the authors state they used the model for passive radiative coolers from Ref. 21. The authors have neglected to mention any details on the atmospheric model or data they used, which is critical to Eq. 3 of Ref. 21. Atmospheric models can vary significantly with geography and season. The authors need to fully specify their model of the atmosphere. I'd further like to draw the authors' attention to a follow-up work [1] by that same group that applied a correction to the atmospheric model in Ref 21.

Comment 3

The authors have neglected to include any experimental uncertainties from their spectrophotometric measurements. These uncertainties must be propagated into their calculations of τ_g , τ_{vis} , τ_{dif} , and ϵ . Without doing so, it is impossible to say if the differences shown in Fig 2G. are statistically meaningful.

Comment 4

The authors should either provide mathematical definitions of their key performance indicators (τ_g , τ_{vis} , τ_{dif} , and ϵ) or cite works which do. It should be clearer how they arrived at each quoted value from the measurements on their spectrophotometers.

Comment 5

Related to the last comment, authors should be clearer about the illumination and light collection geometries of their various spectrophotometric measurements.

Comment 6

The authors have not included any data for wavelengths longer than 16 μm . As thermal radiation is a broadband phenomenon, objects at terrestrial temperatures can have appreciable thermal exchange out to 100 μm or 150 μm . How did the authors account for this?

Comment 7

At 100 % zoom, I find it very difficult to identify τ_{dif} and γ_{dif} in Fig. 2C. If you can extend the vertical scale to start at slightly less than zero, it may be clearer when those values are near zero.

Comment 8

In Figs. 2C and 2D, the change in horizontal scale around 2.5 μm makes it very difficult to see the curves in the 2.5 μm to 5 μm range where they seem to have the most interesting structure.

Comment 9

The caption for Fig. 2E states that transmittance for 60° and 80° is near zero. In this case, I would advise to remove them from the plot and legend to improve clarity.

Comment 10

Should readers assume that the data in Fig. 2F is measured data? Or is it the result of simulations? Additionally, is it for a single wavelength or averaged over some range? Further, having such coarse angular resolution makes it hard to understand the distribution. If it is measured data, the authors used a Cary 7000 UMS module. That instrument's main selling point is that it can be used for repetitive, autonomous measurement. Capturing data at finer angular resolution should not be a burden but it would improve readers' understanding of the transmittance distribution.

Comment 11

Figs. 2G and 3B seem to show some of the same information but Fig. 2G appears to be grouped with experimental data and Fig. 3B appears to be grouped with simulations. It's unclear what is simulated and what is measured.

Comment 12

The authors lean heavily into their metamaterial as a potential building material. Before discussing such benefits as increased photosynthesis rates, it seems prudent to discuss more practical matters such as aging and resiliency to weathering. The effects of heat, humidity, and UV radiation on PDMS have all been investigated previously [2,3]. How would degradation due to aging and weathering impact the performance of this metamaterial?

Comment 13

The authors should revise their more hyperbolic statements to be less subjective. For example:

- Infinite heat sink (line 59)
- Impressive total transmittance (lines 72-73)
- Exceptional transparency (lines 85 and 329)
- Remarkable emissivity (line 86)
- τ_{dif} is exceptionally high (line 174)
- Outstanding emissivity (line 178)
- Exceptional radiative cooling performance (line 265)
- Impressive emissivity (line 332)
- Remarkable aspect (line 334)

Comment 14

There appears to be a small typo on line 176. Should it say 'diffused' instead of 'discussed'?

References

1. Mandal, J., Huang, X. & Raman, A. P. Accurately quantifying clear-sky radiative cooling potentials: A temperature correction to the transmittance-based approximation. *Atmosphere (Basel)* 12, (2021).
2. McIntosh, K. R., Powell, N. E., Norris, A. W., Cotsell, J. N. & Ketola, B. M. The effect of damp-heat and UV aging tests on the optical properties of silicone and EVA encapsulants. *Progress in Photovoltaics: Research and Applications* 19, (2011).
3. Xiu, Y., Zhu, L., Hess, D. & Wong, C. P. Superhydrophobicity and UV stability of polydimethylsiloxane/ polytetrafluoroethylene (PDMS/PTFE) coatings. in *Proceedings of the International Symposium and Exhibition on Advanced Packaging Materials Processes, Properties and Interfaces* vol. 2007 (2007).

Reviewer #4 (Remarks to the Author):

This manuscript reports a surface textured PDMS film integrating high visible diffusion transmittance, radiative cooling and self-cleaning. This film was fabricated by template method, including photolithography, etching, solution casting and peeling. Although interesting, some of main claims of this work are not well supported by the results. The mechanism of the spectral selective and weatherability properties of the film was not stated. Besides, the self-cleaning performance seems not convinced. Therefore, this version cannot be published yet, unless the author can solve the following issues:

1. One of its principal assertions suggests that micron-scale square-based-pyramid surface structures were devised to enhance solar light transmission, particularly in a diffused manner, improve infrared emissivity within the atmospheric window, and attain superhydrophobicity. Nevertheless, there is an absence of a detailed design foundation, including a lack of comprehensive comparative testing for various geometric shapes and parameters. Although the authors do present two PMMM samples with different widths of 6 and 10 μm , this evidence falls short of substantiating the central novelty of this work.
2. Although the authors conducted Ray-tracing simulations to visually depict the light distribution characteristics of the film, these simulations fail to elucidate the connection between the average transmittance, diffused transmittance, and the textured surface's shape. Additionally, the author's attribution of the heightened emissivity to the gradual alteration of the refractive index induced by the pyramid's shape remains vague; it does not sufficiently clarify how the shape of pyramid impacts the intensity and wavelength enhancements.
3. The authors delivered only a superficial analysis apparent in the assessment of surface wettability. The authors showcased the superiority of the surface with 10 μm -wide micro-pyramids over that of 6 μm -wide micro-pyramids. Nonetheless, the manuscript lacks in-depth analysis and discussion in this regard.
4. The authors stated that the superhydrophobic surface can effectively achieve self-cleaning performance, whether through rain in an active manner or dew passively. While it's easy to comprehend how droplets can remove dust, the mechanism for dust removal by dew, as illustrated in Figure 4F, remains unconvincing. I believe that the dust might accumulate in cavities that droplets or dew may not reach, especially in the Cassie model. To bolster this claim, the authors should consider conducting a comparison between the film and glass surfaces.
5. The film thickness seems to exhibit a distinct inconsistency within this study. For instance, in the simulation section, the authors specified the film thickness as 50 μm , whereas Figure 2D depicts the film with a thickness of 70 μm .

Radiative cooling, light management and self-cleaning based on polymer metamaterials

Gan Huang^{1,*}, Ashok R. Yengannagari¹, Kishin Matsumori¹, Prit Patel¹, Anurag Datla¹, Karina Trindade¹, Enkhlen Amarsanaa¹, Tonghan Zhao¹, Uwe Köhler¹, Dmitry Busko¹, Bryce S. Richards^{1,2,*}

Response to Reviewers

Reviewer 1:

This work presents a polymer-based micro-phonic multi-functional metamaterial (PMMM), which integrates several functionalities, including light diffusing, self-cleaning, radiative cooling, while maintaining a high level of transparency. And a detailed discussion of special optical properties (especially the light diffusing in the visible range) is given through theoretical simulations and calculations.

1. In line 141-142, “For applications requiring added strength, the PMMM film can be flattened and adhered to a glass substrate.” How is the adhesion between the PMMM film and glass, and is it able to maintain a stable bond over time? Is it necessary to add additional auxiliary adhesives in the actual construction, and if so, does it affect the optical properties of the material?

Authors' reply: Thank you for highlighting this point. In this study, the PMMM is made of polydimethylsiloxane (PDMS). PDMS naturally exhibits excellent adhesion to glass, a property attributed to the interactions between its siloxane groups (Si–O–Si) and the silanol groups (Si–O–H) present on the glass surface¹. Consequently, additional adhesive is not necessary. We devised two experiments specifically to demonstrate the robust adhesive strength between PDMS and glass, as shown in the additional **Supplementary Fig. 2**.

Supplementary Fig. 2. Adhesion strength testing.

We fabricated a sample by applying a PDMS coating onto a 6-inch glass substrate. Subsequently, this sample was placed outdoors for a duration of two months (14th November 2023 to 15th January 2024)

on campus at the Karlsruhe Institute of Technology in Karlsruhe, Germany. As shown in the **Supplementary Figs. 2a** and **b**, the PDMS-glass interface of the aged sample is subjected to a load using two suction cups to assess its adhesive strength. The PDMS-glass construct can bear a pressure larger than 4 kPa (~ 4.6 kg load on 0.0098 m^2 , as indicated in the figure). Owing to PDMS's renowned stability, the bond it forms with glass remains steadfast, especially when appropriately cured.

In addition, as shown in the **Supplementary Fig. 2c**, we used compressed gas to simulate the strong wind in outdoor applications. The compressed gas generated >100 km/h speed wind (close to the violent storm). The PDMS film on the glass can easily withstand the simulated strong wind.

The above same explanation has been added to the manuscript with red font (Page 4) and also detailed in the Supplementary file.

Reference:

1. Borók, A., Laboda, K., & Bonyár, A. (2021). PDMS bonding technologies for microfluidic applications: A review. *Biosensors*, 11(8), 292.

2. Light management and radiative cooling performance of PMMM are mentioned in this work, but only optical properties and theoretical calculations are available to illustrate its potential application in thermal management, lacking real-world demonstrations or related thermal management experiments. It is recommended to add more content for better visualization of experimental results.

Authors' reply: We appreciate the valuable comments, and we concur that incorporating real-world demonstrations and thermal-management experiments will substantially enrich this study. In response, we have constructed a platform specifically designed to showcase the outdoor performance of PMMM. The outcomes of these experiments have been detailed in the newly added **Fig. 4** as shown below.

Fig. 4. Outdoor testing of radiative cooling performance. **a**, Photograph of the testing platform within the Solar Park of Karlsruhe Institute of Technology. The cooling test chamber was used to test the minimum temperature of

the PMMM sample with radiative cooling. The PMMM-roof box and glass-roof box were used to test the temperature of the experimental enclosures with PMMM and normal glass as the roofs. **b**, Schematics of the cooling test chamber, PMMM-roof box, and glass-roof box. The cooling test chamber was covered with a low-density polyethylene (LDPE) film as the wind shield. Detailed dimensions are provided in the Method section. **c**, PMMM sample temperature (T_{PMMM}), ambient temperature (T_a) and global solar irradiance (G_{sun}). The inserted image shows the sky at 13:00. **d**, Temperature of the PMMM-roof ($T_{\text{PMMM_box}}$) and glass-roof boxes ($T_{\text{glass_box}}$). The inserted image shows the enlarged view of the two boxes.

The outdoor radiative cooling performance of PMMM samples was evaluated at the Solar Park within the Karlsruhe Institute of Technology campus on 7th December 2023, under clear sky conditions, as depicted in **Fig. 4a**. The diagram of the testing platform setup is shown in **Fig. 4b**. All details of the outdoor testing platform setup (dimensions and materials etc.) can be found in the Method section. A cooling test chamber was utilized to test the temperature reduction of the PMMM sample (42.2 cm²) with radiative cooling. This chamber was shielded by a low-density polyethylene (LDPE) film to serve as a wind shield. The chamber's interior was lined with a highly reflective film, while its exterior was insulated thermally. To further assess the cooling efficacy, two boxes were constructed: one with a PMMM roof and the other with a normal glass roof. These boxes were designed to absorb sunlight on their inner walls, allowing for a comparative analysis of the PMMM and glass roofs' cooling performances. The temperatures of PMMM, the boxes, and the ambient were monitored using thermocouples.

The day of testing featured a clear sky as depicted in **Fig. 4c**. The average humidity was approximately 63% between 12:00–15:00. Throughout the test, the temperature of the PMMM was consistently lower than the ambient temperature. At 14:48, the PMMM temperature was ~6 °C lower than the ambient temperature, demonstrating significant cooling in the high-humidity environment of Karlsruhe due to PMMM's strong emissivity. Remarkably, at 16:30, post-sunset, the PMMM temperature dropped to -2.8°C, well below the freezing point, while the ambient temperature was 0.5 °C.

Comparatively, the PMMM-roof box maintained a lower temperature than the glass-roof box until 14:40, attributed to PMMM's superior thermal emissivity, as shown in **Fig. 4d**. At 13:13, the temperature in the PMMM-roof box was 1.7 °C lower than that in the glass-roof box, highlighting the more effective radiative cooling power of the PMMM roof. Interestingly, after 14:40, the temperature in the PMMM-roof box exceeded that of the glass-roof box. This change was due to the micro-pyramid structure's ability to direct sunlight into the box at lower sun elevation angles in the late afternoon (the elevation angle of the sun was ~10° at 14:40), as illustrated by the ray tracing results in **Fig. 3a**. The inserted image offers a view of both the PMMM-roof and glass-roof boxes, showing the PMMM's distinctively blurred and colorful appearance at certain angles.

The above experimental results have been added to the manuscript on Page 10 and Page 11, and all details of the outdoor testing platform setup have been added to the Method section on Page 14:

“Outdoor testing. The cooling test chamber, constructed from polymethyl methacrylate (PMMA), features a 5-cm-thick layer of polyethylene foam with an aluminum foil coating on its outer wall. This layer is specifically designed to minimize heat transfer with the surrounding environment. The chamber is also equipped with a low-density polyethylene (LDPE) foil as a wind shield. Its inner wall is lined with high-reflective reflectors (*Alanod*) to enhance its insulation properties. Both the PMMM-roof box and the glass-roof box are similarly constructed from PMMA and are coated with aluminum reflector foil. Their inner walls are lined with solar absorber foil (*Alanod*). Each sample measured in the experiment has an area of 42.2 cm². The internal dimensions of the cooling test chamber are 100

mm × 100 mm × 20 mm, while the PMMM-roof box and glass-roof box measure 65 mm × 65 mm × 100 mm each. Temperature measurements were conducted using thermocouples (*Pico*). The ambient temperature was measured by a thermocouple placed in the air near the test platform. The temperature of the PMMM was measured by attaching a thermocouple beneath it. The thermocouple was then subjected to pressure from a spring, as depicted in Figure 4b. The entire testing platform was positioned horizontally. Solar irradiance was monitored using a pyranometer (*Eppley*), wind speed was gauged with a cup anemometer (*BTMETER*), and relative humidity was recorded using a hygrometer (*ThermoPro*).”

3. Figure 2E and F discuss the diffusion angle of transmitted light, which is mainly located within 30 degrees, so is there any connection between this angle and the surface microstructure of PMMM? And is it possible to further increase the diffusion angle of transmitted light, so that the sunlight can be more fully utilized, which will make the indoor lighting environment friendlier?

Authors’ reply: Yes, there is indeed a direct relationship between the diffusion angle (or the “refraction angle”) and the surface microstructure of PMMM. When light interacts with the PMMM, it is refracted by the micro-pyramid structure and redirected in specific directions. This diffusion angle largely depends on the base angle of the micro-pyramids. For instance, when the base angle is set at 54.7° in this study, the light exits the bottom substrate at approximately 27°, aligning with the experimental diffusion angle observed (refer to additional simulation results in **Supplementary Fig. 6**, and experimental results in **Fig. 2f**).

Further, we believe it is feasible to alter the refraction angle by adjusting the base angle of the micro-pyramids. To validate this hypothesis, we conducted additional analyses and simulations by using the validated numerical model, the results of which are presented in **Supplementary Fig. 6**. Our analysis, grounded in Snell’s law, indicates that the diffusion angle increases as the micro-pyramid’s base angle enlarges. The simulation results show that the maximum diffusion angle (φ) expands with an increasing base angle (θ_b) of the micro-pyramid. As depicted in **Supplementary Figs. 6b and 6c**, the diffusion angle broadens from 18° to 41° when the micro-pyramid’s base angles range from 40° to 70°. This suggests potential for enhancing the utilization of sunlight in indoor environments, thereby creating a friendlier lighting atmosphere. We have included this explanation in the main text on Page 6.

Supplementary Fig. 6. Correlation between the diffusion angle and the base angle of the micro-pyramid.

4. The “Introduction” and “Discussion” sections are too lengthy and could be streamlined to improve readability.

Authors’ reply: Thank you for the comments. We have deleted some unnecessary sentences in the Introduction and Discussion sections (around 10 rows of text have been deleted). The word count of “Introduction” has been reduced from 941 to 894. The word count of “Discussion” has been reduced from 278 to 196.

5. The various optical property indexes mentioned in **Figure 2G**, do they refer to the value at a certain wavelength or the average value in the spectral range? If it involves the calculation of average values (e.g., the average infrared emissivity at 8-13 μm), the relevant formulas and the calculation process should be elaborated in detail in the “Methods” section.

Authors’ reply: In **Fig. 2g**, the global transmittance τ_{g_ave} and diffuse transmittance τ_{dif_ave} are the spectral weighted values in the range of 0.3 to 2.5 μm for the air-mass 1.5 (AM1.5) standard solar spectrum; the global visible transmittance τ_{vis_ave} is the spectrally-weighted value in the visible range of 0.38 to 0.7 μm ; the emissivity ε_{ave} is the average value in the range of 8 to 13 μm . The below equations are used to calculate the values of τ_{g_ave} , τ_{dif_ave} , τ_{vis_ave} and ε_{ave} :

$$\tau_{g_ave} = \int_{0.3 \mu\text{m}}^{2.5 \mu\text{m}} \tau_g(\lambda) G_{AM1.5}(\lambda) d\lambda / I_{AM1.5} \quad (1)$$

$$\tau_{dif_ave} = \int_{0.3 \mu\text{m}}^{2.5 \mu\text{m}} \tau_{dif}(\lambda) G_{AM1.5}(\lambda) d\lambda / I_{AM1.5} \quad (2)$$

$$\tau_{vis_ave} = \int_{0.38 \mu\text{m}}^{0.7 \mu\text{m}} \tau_g(\lambda) G_{AM1.5}(\lambda) d\lambda / \int_{0.38 \mu\text{m}}^{0.7 \mu\text{m}} G_{AM1.5}(\lambda) d\lambda \quad (3)$$

$$\varepsilon_{ave} = \int_{8 \mu\text{m}}^{13 \mu\text{m}} \varepsilon(\lambda) d\lambda / (13 - 8 \mu\text{m}) \quad (4)$$

where $I_{AM1.5}$ is the solar irradiance of AM1.5 standard solar spectrum, and $G_{AM1.5}$ is AM1.5 standard solar spectrum.

The above formulas and the calculation process have been added the “Method” section. We discovered an error in the previous calculation on the values of transmittances. We have carefully recalculated and updated all the data in **Fig. 2g** according to the above formulae.

Reviewer 2:

The authors propose a polymer-based micro-photonic multi-functional metamaterial (PMMM) film of the enhanced radiative cooling characteristics and the anti-dusting feature. Microscale pyramid structures are proposed for the better radiative cooling features. In addition, the self-cleaning properties are shown upon the anti-adhesive coating on the microscale surface structures. The work is interesting, and the manuscript is well-written. However, the following comments should be addressed for further consideration of accepting this manuscript to Nature Communications.

1. In Fig. 3, the ray propagation maps of the PMMA in terms of is interesting like the omnidirectional antireflection properties of moth-eye structures. From that perspective, the structured film seems to exhibit excellent radiative cooling performance upon the simulated incident rays for a day, compared to planar thin film. The authors should provide the estimation on the time-integrated radiative cooling performance upon the simulated incident rays for a day, compared to the planar film to highlight the figure-of-merits of the PMMM film.

Authors' reply: We sincerely appreciate the suggestion and acknowledge its significance in demonstrating the advantages of the PMMM film. While theoretical estimations can offer insights, we believe that experimental evidence provides a more compelling and accurate representation of performance. Therefore, in line with the reviewer's recommendation, we opted to conduct outdoor experiments that reflect the real-world conditions and performance of our PMMM film. We hope that the experimental data obtained from these outdoor tests will effectively demonstrate the enhanced performance and advantages of the PMMM film in practical applications. To this end, we have included an additional figure (Fig. 4) that details the outdoor experiment setup and findings. These tests were carried out over the course of a day, allowing us to directly compare the cooling performance of the PMMM film with that of a standard planar film.

Fig. 4. Outdoor testing of radiative cooling performance. a, Photograph of the testing platform within the Solar Park of Karlsruhe Institute of Technology. The cooling test chamber was used to test the minimum temperature of the PMMM sample with radiative cooling. The PMMM-roof box and glass-roof box were used to test the

temperature of the experimental enclosures with PMMM and normal glass as the roofs. **b**, Schematics of the cooling test chamber, PMMM-roof box, and glass-roof box. The cooling test chamber was covered with a low-density polyethylene (LDPE) film as the wind shield. Detailed dimensions are provided in the Method section. **c**, PMMM sample temperature (T_{PMMM}), ambient temperature (T_a) and global solar irradiance (G_{sun}). The inserted image shows the sky at 13:00. **d**, Temperature of the PMMM-roof ($T_{\text{PMMM_box}}$) and glass-roof boxes ($T_{\text{glass_box}}$). The inserted image shows the enlarged view of the two boxes.

The outdoor radiative cooling performance of PMMM samples was evaluated at the Solar Park within the Karlsruhe Institute of Technology campus on 7th December 2023, under clear sky conditions, as depicted in **Fig. 4a**. The diagram of the testing platform setup is shown in **Fig. 4b**. All details of the outdoor testing platform setup (dimensions and materials etc.) can be found in the Method section. A cooling test chamber was utilized to test the temperature reduction of the PMMM sample (42.2 cm²) with radiative cooling. This chamber was shielded by a low-density polyethylene (LDPE) film to serve as a wind shield. The chamber's interior was lined with a highly reflective film, while its exterior was insulated thermally. To further assess the cooling efficacy, two boxes were constructed: one with a PMMM roof and the other with a normal glass roof. These boxes were designed to absorb sunlight on their inner walls, allowing for a comparative analysis of the PMMM and glass roofs' cooling performances. The temperatures of PMMM, the boxes, and the ambient were monitored using thermocouples.

The day of testing featured a clear sky as depicted in **Fig. 4c**. The average humidity was approximately 63% between 12:00–15:00. Throughout the test, the temperature of the PMMM was consistently lower than the ambient temperature. At 14:48, the PMMM temperature was ~6 °C lower than the ambient temperature, demonstrating significant cooling in the high-humidity environment of Karlsruhe due to PMMM's strong emissivity. Remarkably, at 16:30, post-sunset, the PMMM temperature dropped to -2.8°C, well below the freezing point, while the ambient temperature was 0.5 °C.

Comparatively, the PMMM-roof box maintained a lower temperature than the glass-roof box until 14:40, attributed to PMMM's superior thermal emissivity, as shown in **Fig. 4d**. At 13:13, the temperature in the PMMM-roof box was 1.7 °C lower than that in the glass-roof box, highlighting the more effective radiative cooling power of the PMMM roof. Interestingly, after 14:40, the temperature in the PMMM-roof box exceeded that of the glass-roof box. This change was due to the micro-pyramid structure's ability to direct sunlight into the box at lower sun elevation angles in the late afternoon (the elevation angle of the sun was ~10° at 14:40), as illustrated by the ray tracing results in **Fig. 3a**. The inserted image offers a view of both the PMMM-roof and glass-roof boxes, showing the PMMM's distinctively blurred and colorful appearance at certain angles.

The above experimental results have been added to the manuscript on Page 10, and all details of the outdoor testing platform setup have been added to the Method section on Page 14:

“Outdoor testing. The cooling test chamber, constructed from polymethyl methacrylate (PMMA), features a 5-cm-thick layer of polyethylene foam with an aluminum foil coating on its outer wall. This layer is specifically designed to minimize heat transfer with the surrounding environment. The chamber is also equipped with a low-density polyethylene (LDPE) foil as a wind shield. Its inner wall is lined with high-reflective reflectors (*Alanod*) to enhance its insulation properties. Both the PMMM-roof box and the glass-roof box are similarly constructed from PMMA and are coated with aluminum reflector foil. Their inner walls are lined with solar absorber foil (*Alanod*). Each sample measured in the experiment has an area of 42.2 cm². The internal dimensions of the cooling test chamber are 100 mm × 100 mm × 20 mm, while the PMMM-roof box and glass-roof box measure 65 mm × 65 mm ×

100 mm each. Temperature measurements were conducted using thermocouples (*Pico*). The ambient temperature was measured by a thermocouple placed in the air near the test platform. The temperature of the PMMM was measured by attaching a thermocouple beneath it. The thermocouple was then subjected to pressure from a spring, as depicted in Figure 4b. The entire testing platform was positioned horizontally. Solar irradiance was monitored using a pyranometer (*Eppley*), wind speed was gauged with a cup anemometer (*BTMETER*), and relative humidity was recorded using a hygrometer (*ThermoPro*).”

2. The radiative cooling power of the proposed PMMM film is just simulated. The experimental results must be accompanied to the simulation results. To support the claims on the structural merits, the experimental results on the radiative cooling are very important, but it is missing.

Authors’ reply: Thank you for the comments. We agree with the significance of providing experimental results for radiative cooling. As mentioned in our response to Comment 1, we have incorporated outdoor experiments to demonstrate the real-world performance of our study. The results of these outdoor experiments, focusing on radiative cooling, have been added to the main manuscript. Due to the unavailability of atmospheric transmittance data for Karlsruhe in accessible databases, we were unable to perform simulations for outdoor performance in this specific location and compare with the experimental data. Nevertheless, we are confident that the experimental data presented will convincingly demonstrate the efficacy of radiative cooling in real-world conditions.

For **Figs. 3e** and **3f**, we simulated the radiative cooling power of PMMM using atmospheric data as provided by MODTRAN. This simulation approach to estimating the PMMM film’s radiative cooling performance is consistent with widely recognized numerical methods in the literature (referencing *Raman et al., Nature, 515, pp. 540–544, 2014*). Furthermore, to validate our simulation model, we compared it against the experimental results from *Raman et al.*, as shown in another **Supplementary Fig. 15b**. We found that our simulation results align closely with these experimental findings.

The above discussion has been added to the manuscript on Page 9 and also the supplementary file.

Supplementary Fig. 15. a, Atmospheric transmittance for the modelling. **b,** Simulation results compared to experimental results. The experiment was conducted by *Raman et al.* in Stanford (*Raman et al., Nature, 515, pp. 540–544, 2014*), California, in mid-December 2013. The heat convection coefficient was calculated by $h_{wind} = 2.5 + 2 \cdot v_{wind}$, where v_{wind} is the wind speed (*Zhao et al., Joule, 3(1), pp. 111-123, 2019*).

3. The authors estimated the enhancement of photosynthesis upon the deployment of the PMMA film by ~9%. The enhancement can be attributed to the importance of diffused light for photosynthesis based on the reference. When the authors more elaborate the discussions in terms of the importance of diffused light in some specific applications, it would be helpful to the readers to understand the importance of this study.

Authors' reply: We appreciate the reviewer's insightful comments and agree that a discussion on the broader implications of diffused light will enrich the manuscript and help readers appreciate the significance of our study.

Diffused light is indeed crucial for optimizing photosynthesis, as it enables a more uniform light distribution, allowing plants to utilize light more effectively for growth. Beyond agriculture, the advantages of diffused light extend to several domains, enhancing both functional and aesthetic aspects of various environments. In architectural applications, for instance, diffused light mitigates glare and provides balanced indoor illumination, which contributes to visual comfort and reduces eye strain. This even light distribution is beneficial in workplaces, educational settings, and healthcare facilities, where it supports human performance and well-being. Moreover, diffused light can enhance privacy by softening and scattering direct views through transparent media, an aspect particularly relevant to our polymer-based micro-photonic multi-function metamaterial (PMMM). The high diffusion transmittance of the PMMM film not only serves aesthetic purposes but also promotes privacy and creates a sense of openness without compromising on light quality or energy efficiency. In retail and exhibition spaces, such as art galleries and museums, diffused light plays a pivotal role in presenting products and artworks in a way that minimizes reflections and hotspots, ensuring that each item is viewed under optimal lighting conditions.

To underscore these points, we have expanded our discussion in the manuscript to illustrate the multifaceted applications of diffused light, thereby underlining the versatility and transformative potential of the PMMM film. We believe that these additions will provide the readers with a comprehensive understanding of the importance of diffused light and the wide-reaching impact of our research in this field. The above analysis has been added to the manuscript on page 9.

4. For anti-dusting characteristics, the information, including size and key element distributions of dusts, on the tested dust particles should be provided, together with the discussions on the mechanism of anti-dusting, for the better understanding and the further optimization of self-cleaning and anti-dusting properties.

Authors' reply: We thank the reviewer for the valuable feedback emphasizing the importance of providing a detailed characterization of the dust particles used in our self-cleaning tests. Such information is indeed critical to validate the anti-dusting performance of our PMMM film and to elucidate the underlying mechanism of its self-cleaning properties.

In response to this insightful comment, we have now included a comprehensive analysis of the dust utilized in our experiments (the real dust particles were collected from a local window in Karlsruhe, Germany). We have characterized the dust particles in terms of their size distribution, shape, and elemental composition using scanning electron microscopy (SEM) and energy-dispersive X-ray spectroscopy (EDX), as shown in **Supplementary Fig. 19**, which is reproduced below for the easy reference of the reviewer. The SEM images in **Supplementary Fig. 19a to c** shows that the dust

particle sizes predominantly larger than 10 μm , with a majority being silicate and carbonate minerals, which are common constituents of atmospheric dust.

Supplementary Fig. 19. SEM/EDX characterization of the dust.

The anti-dusting capabilities are the result of a synergy between the surface microstructure and the inherent material properties. The PMMM film is textured with micro-pyramid features that replicate the hierarchical micro- and nano-scale roughness found in superhydrophobic natural surfaces, like lotus leaves. This structure reduces the contact area between dust particles and the surface, significantly lowering the adhesion force due to the Cassie-Baxter wetting state, where air pockets are trapped underneath water droplets. The polydimethylsiloxane used in the PMMM provides a low surface energy, which inherently repels water and thus, aqueous-based dirt and dust particles. This hydrophobic nature is further enhanced by the micro-pyramid surface patterning, contributing to a superhydrophobic surface with a high contact angle for water droplets. When water droplets impact or roll across the PMMM surface, they collect dust particles through adhesion and momentum transfer. The water droplets, with the entrapped dust, then readily roll off due to the low adhesion, effectively cleaning the surface. This process is akin to the "lotus effect," where rainwater washes away contaminants.

We have added discussions around these findings to elucidate how the unique surface morphology of the PMMM contributes to its self-cleaning capabilities. By analyzing the interaction between the surface and the dust particles, we have gained insights that will inform future optimization of the PMMM film for even better self-cleaning and anti-dusting properties.

The above analysis has been added to the manuscript on page 11 and also the supplementary file.

5. Finally, wafer-scale fabrication of the PMMM film was successfully achieved, but for the real application on the roof, much larger-scale fabrication is required. Further developments should be discussed in detailed manner for practical aspects of the PMMM film.

Authors' reply: To facilitate the scaling up of PMMM material for practical applications, we have devised two potential methods. The first approach, illustrated in **Supplementary Fig. 4a**, involves the straightforward assembly of a matrix of small silicon wafer moulds. This stitching technique allows for the creation of a larger-sized module suitable for the production of extensive PMMM films. **Supplementary Fig. 4b** features an example where four small silicon pieces ($6.5\text{ cm} \times 6.5\text{ cm}$) are combined to form a larger module ($13\text{ cm} \times 13\text{ cm}$), with a 1 Euro coin provided for scale reference. The second method, depicted in **Supplementary Fig. 4c**, employs hot-embossing technology for the fabrication of large-scale PMMM films. Hot embossing is renowned for its ability to consistently produce high-quality microstructures across expansive areas. This technique's precision and scalability make it an ideal choice for producing PMMM films on a larger scale.

Supplementary Fig. 4. Potential ways to upscale the PMMM material.

We have added the above methods in the main text on Page 4 and the Supplementary file, to provide detailed guidance of material upscaling for future work.

Reviewer 3:

In their manuscript "Radiative cooling, light management and self-cleaning based on polymer metamaterials," the authors propose a polymer-based micro-phonic multi-functional metamaterial (PMMM) as a novel building material for transparent roofs and walls. Their PMMM exhibits high, diffuse transmittance to visible wavelengths and high infrared emittance in the atmospheric transmission window. Further, the microstructure of the PMMM results in a superhydrophobic surface, enhancing its self-cleaning properties. All-in-all, the advantages of such a material sound promising as an alternative building material. While I believe the work would be appropriate and interesting to the community of Nature Communications readers, the manuscript itself needs revision before it is acceptable for publication. To that end, I have included below 14 comments which should be addressed before this work is ready for release. I am optimistic that the authors can satisfactorily address my concerns and communicate their excellent science but, until that time, I cannot recommend publication.

1. One of the most important functions of more traditional, opaque passive radiative coolers is to reject as much incoming solar radiation as possible. This is due to the sheer magnitude of the incoming solar irradiance compared to how much heat a blackbody can emit at terrestrial temperatures. In the case of this metamaterial, transparency is valued since the sunlight can be used to offset lighting needs. Relative to glass, the metamaterial transmits more solar radiation into the structure. That means more solar energy will eventually be thermalized in the structure, increasing the cooling needs of the building. The authors mention the problem of excess heat accumulation starting on line 39, but then promote a technology that, at first glance, would seem to make the problem worse. I would like to see a discussion by the authors addressing this. Is there an optimal transmittance to balance cooling and lighting energy usage, or is it always better to transmit as much or as little as possible?

Authors' reply: Thank you for recognizing the importance of our metamaterial's transparency. You raise a pertinent point about the potential trade-off between light transmittance and heat accumulation. Based on our data, the transmittances for glass and the metamaterial are 0.92 and 0.95 respectively, while their emissivity are 0.86 and 0.98. Although the metamaterial transmits slightly more solar radiation (a relatively 3.2% increase) compared to glass, its significantly higher emissivity (a 13.9% relative increase) effectively enhances thermal radiation. Consequently, despite attracting marginally more sunlight, the metamaterial can still potentially reduce a building's cooling demands, especially in conditions where radiative cooling is predominant. Our additional experimental results, using two boxes with PMMM and glass roofs, corroborate this. As shown in **Fig. 4d** (reproduced below), the PMMM-roof box consistently exhibited lower temperatures compared to the glass-roof box, owing to PMMM's superior thermal emissivity. For instance, at 13:13, the temperature in the PMMM-roof box was 1.7 °C lower than in the glass-roof box, demonstrating PMMM's more effective radiative cooling, despite its higher transmittance.

Fig. 4d, Temperature of the PMMM-roof box ($T_{\text{PMMM_box}}$) and glass-roof box ($T_{\text{glass_box}}$). The inserted image shows the photographs of the two boxes.

We appreciate the reviewer’s inquiry regarding the optimal transmittance balance for cooling and lighting energy usage. Indeed, this topic is characterized by its complexity and the absence of a universal solution. The determination of an 'optimal' transmittance level is intricately linked to a multitude of variables. These include, but are not limited to, the specific requirements for cooling and illumination, the prevailing environmental conditions like solar irradiance and ambient temperature, as well as the architectural characteristics of the building in question.

It is our understanding that each scenario demands a unique approach and additional comprehensive research, which is out of the main scope of this research. For instance, in situations where natural lighting is a priority, a higher transmittance may prove advantageous. Conversely, in contexts where the reduction of cooling loads is paramount, a lower transmittance might be more appropriate. While our current findings offer foundational insights on the metamaterial, we acknowledge the necessity for ongoing research. Future studies, we hope, will further refine these models and contribute significantly to the development of more energy-efficient buildings.

We have added the above discussion on Page 10.

2. On line 247, the authors state they used the model for passive radiative coolers from Ref. 21. The authors have neglected to mention any details on the atmospheric model or data they used, which is critical to Eq. 3 of Ref. 21. Atmospheric models can vary significantly with geography and season. The authors need to fully specify their model of the atmosphere. I’d further like to draw the authors’ attention to a follow-up work [1] by that same group that applied a correction to the atmospheric model in Ref 21.

[1] Mandal, J., Huang, X. & Raman, A. P. Accurately quantifying clear-sky radiative cooling potentials: A temperature correction to the transmittance-based approximation. *Atmosphere (Basel)* 12, (2021).

Authors’ reply: We appreciate the reviewer’s comment and acknowledge the importance of detailing the atmospheric model employed. We are also grateful for the information regarding the updated model by Raman’s group. In the revised version, we have revised our methodology to incorporate the corrected atmospheric model (Mandal *et al.*, *Atmosphere*, 12, 2021) to the previous model in Ref. 21 (especially revised the way of calculating the radiation of atmosphere in the Eq. 3 of Ref. 21). To provide clarity, we have provided details of the modelling in the Method section:

“Radiative cooling power simulation model. In our study, we adopted the energy balance model initially proposed by Raman et al. (Ref. 21) to simulate radiative cooling power. This model is widely recognized for its effectiveness in calculating the potential radiative cooling power under specific atmospheric conditions, specifically the MODTRAN Standard Atmosphere used in Ref. 21 and also this study, characterized by midlatitude, clear skies, and low humidity. Notably, we have integrated a refined atmospheric model recently developed by Raman’s group (Ref. 58) to enhance the accuracy of our calculations, particularly regarding atmospheric thermal radiation. The updated model distinguishes between two primary atmospheric contributors: Ozone, located higher in the atmosphere, and a combination of CO₂ and water vapor, which is distributed throughout the atmosphere. This differentiation is reflected in the formula⁵⁸:

$$I_{\text{atm}}(\theta, \lambda) = \varepsilon_{\text{Ozone}}(\theta, \lambda) \cdot I_{\text{bb}}(T_{\text{Ozone}}, \lambda) + \varepsilon_{\text{rest}}(\theta, \lambda) \cdot I_{\text{bb}}(T_{\text{rest}}, \lambda) \quad (5)$$

where $\varepsilon_{\text{Ozone}}$ and T_{Ozone} are the emissivity and temperature of the Ozone layer; $\varepsilon_{\text{rest}}$ and T_{rest} are the emissivity and the temperature of the rest of the atmospheric components; I_{bb} is the thermal irradiance of the black body. The detailed raw data utilized for this corrected atmospheric model is accessible in Reference 58.”

Correspondingly, we have updated the simulation results in **Figs. 3e** and **3f**, as well as the related data within the manuscript.

3. The authors have neglected to include any experimental uncertainties from their spectrophotometric measurements. These uncertainties must be propagated into their calculations of τ_{g} , τ_{vis} , τ_{dif} , and ε . Without doing so, it is impossible to say if the differences shown in Fig 2G. are statistically meaningful.

Authors’s reply: Thank you for the insightful comment regarding the inclusion of experimental uncertainties in our spectrophotometric measurements. We would like to clarify that the Agilent Cary 7000 spectrophotometer, which we utilized for our measurements, has a very low inherent measurement error post-calibration, typically within 0.01%. Despite this high accuracy, we recognize that the primary source of uncertainty in our experiments is not from the instrument itself but rather from how the samples are positioned within it. This uncertainty arises because even minor variations in the placement of the samples can affect the path and angle of light as it interacts with the sample, leading to slight variations in the measurement results. Small deviations in the angle or position can change the way light is scattered, reflected, or absorbed, hence impacting the spectrophotometric readings. To address this concern, we have re-evaluated the optical properties and emissivity of our samples. This involved conducting 10 separate measurements, removing and then remounting the sample in the device each time. The additional **Supplementary Figs. 7a** and **7b** displays the variations in transmittance and emissivity observed across these 10 measurements. Further, we have compiled the average transmittance values in the 0.3–2.5 μm range from these measurements in the additional **Supplementary Fig. 7c**. This data indicates an uncertainty margin of 1%. Similarly, the average emissivity in the 8–13 μm range, derived from the Bruker Vertex 70 Fourier-transform infrared spectroscopy, is presented in **Supplementary Fig. 7d**. Here, the observed uncertainty is as low as 0.2%. We have also added the errors of the measurements in the main text.

Supplementary Fig. 7. Error of 10-times repeated measurements.

4. The authors should either provide mathematical definitions of their key performance indicators (τ_g , τ_{vis} , τ_{dif} , and ε) or cite works which do. It should be clearer how they arrived at each quoted value from the measurements on their spectrophotometers.

Authors' reply: In the **Fig. 2g**, the global transmittance τ_{g_ave} and diffuse transmittance τ_{dif_ave} are the spectral weighted values in the range of 0.3 to 2.5 μm for the air-mass 1.5 (AM1.5) standard solar spectrum; the global visible transmittance τ_{vis_ave} is the spectrally-weighted value in the visible range of 0.38 to 0.7 μm ; the emissivity ε_{ave} is the average value in the range of 8 to 13 μm . The below equations are used to calculate the values of τ_{g_ave} , τ_{dif_ave} , τ_{vis_ave} and ε_{ave} :

$$\tau_{g_ave} = \int_{0.3 \mu\text{m}}^{2.5 \mu\text{m}} \tau_g(\lambda) G_{AM1.5}(\lambda) d\lambda / I_{AM1.5} \quad (1)$$

$$\tau_{dif_ave} = \int_{0.3 \mu\text{m}}^{2.5 \mu\text{m}} \tau_{dif}(\lambda) G_{AM1.5}(\lambda) d\lambda / I_{AM1.5} \quad (2)$$

$$\tau_{vis_ave} = \int_{0.38 \mu\text{m}}^{0.7 \mu\text{m}} \tau_g(\lambda) G_{AM1.5}(\lambda) d\lambda / \int_{0.38 \mu\text{m}}^{0.7 \mu\text{m}} G_{AM1.5}(\lambda) d\lambda \quad (3)$$

$$\varepsilon_{ave} = \int_{8 \mu\text{m}}^{13 \mu\text{m}} \varepsilon(\lambda) d\lambda / (13 - 8 \mu\text{m}) \quad (4)$$

where $I_{AM1.5}$ is the solar irradiance of AM1.5 standard solar spectrum, and $G_{AM1.5}$ is AM1.5 standard solar spectrum.

The above formulas and the calculation process have been added the ‘‘Method’’ section. We discovered an error in the previous calculation on the values of transmittances. We have carefully re-calculated and updated all the data in **Fig. 2g** according to the above formulae.

The above formulas and the calculation process have been added the ‘‘Method’’ section. We inspected an error in the previous calculation. We have carefully re-calculated and updated all the data in **Fig. 2g** according to the above formulas and updated.

5. Related to the last comment, authors should be clearer about the illumination and light collection geometries of their various spectrophotometric measurements.

Authors' reply: Thanks for the comment. To provide a clearer understanding of our measurement approach, we have included a supplementary figure that illustrates the illumination and light collection geometries used in our spectrophotometric measurements. **Supplementary Fig. 5** visually represents the top view geometries for different types of measurements: (a) global transmittance, (b) global reflectance, (c) diffuse transmittance, and (d) diffuse reflectance. In these setups, either direct or diffuse light is captured using an integration sphere, which features a highly reflective internal surface. This light is then directed to detectors located at the bottom of the sphere for accurate measurement. Prior to conducting these measurements, the spectrophotometer is meticulously calibrated to ensure precision and reliability in our results.

We have added the above discussion on Page 5, and also to the supplementary file.

Supplementary Fig. 5. Illumination and light collection geometries (top view). **a**, global transmittance. **b**, global reflectance. **c**, diffuse transmittance. **d**, diffuse reflectance.

6. The authors have not included any data for wavelengths longer than 16 μm . As thermal radiation is a broadband phenomenon, objects at terrestrial temperatures can have appreciable thermal exchange out to 100 μm or 150 μm . How did the authors account for this?

Authors' reply: Thank you for the insightful comment. To clearly present the material's emissivity around the transmittance window (8–13 μm), we thus have limited the data displayed in our figures to within 16 μm . We concur with the reviewer's observation that thermal radiation exchange between terrestrial objects remains significant beyond the 16 μm range. As demonstrated in the additional **Supplementary Fig. 16a**, the thermal emission from a blackbody at temperatures of 300 K and 350 K predominantly occurs within the 3–50 μm wavelength range. To address this, **Supplementary Fig. 16b** has been included to show the emissivity of our sample (PMMM with 10- μm micro-pyramids) over this extended wavelength range. In our numerical model used for calculating radiative cooling performance, we had incorporated considerations of thermal radiation exchange within the 3–50 μm range.

We have added the above discussion on Page 9, and also to the supplementary file.

Supplementary Fig. 16. **a**, Thermal radiation spectrum of black body. **b**, Emissivity of PMMM

7. At 100 % zoom, I find it very difficult to identify τ_{dif} and γ_{dif} in Fig. 2C. If you can extend the vertical scale to start at slightly less than zero, it may be clearer when those values are near zero.

Authors' reply: Thanks for pointing out this issue. The vertical scale of **Fig. 2c** has now been slightly extended to start from -0.01, and τ_{dif} and γ_{dif} can be clearer now. To be consistent, the vertical scale of Fig. 2d has also been adjusted to be the same as that of **Fig. 2c**.

8. In Figs. 2C and 2D, the change in horizontal scale around 2.5 μm makes it very difficult to see the curves in the 2.5 μm to 5 μm range where they seem to have the most interesting structure.

Authors' reply: Thank you for the comment. To improve the visibility and clarity of the data, particularly in the mentioned range, we have eliminated the break in the horizontal axis at 2.5 μm . Additionally, we have adopted a logarithmic scale for the x-axis. These adjustments aim to provide a clearer and more detailed view of the curves within the 2.5 μm to 5 μm range. We hope that these changes make the figures more informative.

9. The caption for Fig. 2E states that transmittance for 60° and 80° is near zero. In this case, I would advise to remove them from the plot and legend to improve clarity.

Authors' reply: Thanks for the comment –the transmittance curves for 60° and 80° have now been removed.

10. Should readers assume that the data in Fig. 2F is measured data? Or is it the result of simulations? Additionally, is it for a single wavelength or averaged over some range? Further, having such coarse angular resolution makes it hard to understand the distribution. If it is measured data, the authors used a Cary 7000 UMS module. That instrument's main selling point is that it can be used for repetitive, autonomous measurement. Capturing data at finer angular resolution should not be a burden but it would improve readers' understanding of the transmittance distribution.

Authors' reply: The data presented in **Fig. 2f** represents the polar transmittance distribution of the PMMM sample at a 500 nm wavelength, as measured by the Agilent Cary 7000 UMS module. We have updated the figure caption to explicitly state that these are experimental results. In response to the reviewer's suggestion for improved visualization, we have enhanced the resolution of the updated **Fig. 2f** from 10° to a much more precise 1°. This modification allows for a more detailed representation of the transmittance distribution.

Additionally, we have increased the measurement resolution for **Fig. 2e**, aiming to present a clearer view of the spectral transmittance distribution. While this enhancement was not specifically requested by the reviewer, we acknowledge its value in deepening our understanding of the transmittance distribution patterns.

Fig. 2. Angular transmittance distributions

11. Figs. 2G and 3B seem to show some of the same information but Fig. 2G appears to be grouped with experimental data and Fig. 3B appears to be grouped with simulations. It's unclear what is simulated and what is measured.

Authors' reply: Both the **Figs. 2g** and **3b** are measured results. We have revised the captions of both figures to clearly clarify in the figures is measured or simulated, to avoid misunderstanding.

12. The authors lean heavily into their metamaterial as a potential building material. Before discussing such benefits as increased photosynthesis rates, it seems prudent to discuss more practical matters such as aging and resiliency to weathering. The effects of heat, humidity, and UV radiation on PDMS have all been investigated previously [2,3]. How would degradation due to aging and weathering impact the performance of this metamaterial?

Authors' reply: We appreciate the reviewer's emphasis on the practicality of our metamaterial film, especially concerning its durability and resistance to aging and weathering in real-world applications. As highlighted in Reference 2, the transmittance of PDMS—a primary constituent of our metamaterial—exhibits minimal reduction (only by 0.14%) in damp-heat accelerated aging tests (1325 hours). Furthermore, UV radiation appears to have a negligible impact on its optical properties. Based on this, we anticipate that our metamaterial's stability will mirror that of PDMS, given that PDMS forms the sole base material of the metamaterial.

To provide a more comprehensive assessment of our metamaterial's stability, we conducted a real-world test by exposing samples to the outdoor environment in Karlsruhe, Germany, for durations of one and two months. Specifically, the 1-month sample was exposed from 14th November 2023 to 15th December 2023, while the 2-month sample was left outdoors from 14th November 2023 to 15th January 2024. Post-exposure, we re-evaluated the metamaterial's optical performance. The results, which are elaborated in **Supplementary Figs. 21a** and **b**, show a transmittance reduction of 3% and emissivity reduction of 1% after one month. It's important to note that both transmittance and emissivity underwent only marginal changes with the extension of exposure time to two months. This data suggests a relative stability of the PDMS-based metamaterial under extended outdoor environmental conditions.

Supplementary Fig. 21. Degradation of PMMM in outdoor testing.

The performance of both the 1-month and 2-month samples, as depicted in **Supplementary Figs. 21a** and **b**, was assessed directly post-exposure, without any cleaning, upon their return from the outdoor environment. Therefore, the observed changes in performance can be attributed to a combination of aging and soiling effects. To disentangle the impacts of aging from soiling, the 2-month sample underwent a cleaning process using distilled water. We then reassessed its transmittance and emissivity for a comparative analysis with the pre-cleaned state, as shown in **Supplementary Figs. 21c** and **d**. The results revealed negligible changes in both transmittance and emissivity post-cleaning, suggesting that aging, rather than soiling, is the primary factor contributing to the slight reductions observed in these optical properties.

We have added the above discussion on Page 13, and also to the supplementary file.

13. The authors should revise their more hyperbolic statements to be less subjective. For example:

- Infinite heat sink (line 59)
- Impressive total transmittance (lines 72-73)
- Exceptional transparency (lines 85 and 329)
- Remarkable emissivity (line 86)
- τ_{dif} is exceptionally high (line 174)
- Outstanding emissivity (line 178)
- Exceptional radiative cooling performance (line 265)
- Impressive emissivity (line 332)
- Remarkable aspect (line 334)

Authors' reply: We have carefully gone through the manuscript and removed all such statements.

14. There appears to be a small typo on line 176. Should it say 'diffused' instead of 'discussed'?

Authors' reply: Thank you for pointing out this. We have revised it.

References

1. Mandal, J., Huang, X. & Raman, A. P. Accurately quantifying clear-sky radiative cooling potentials: A temperature correction to the transmittance-based approximation. *Atmosphere (Basel)* 12, (2021).
2. McIntosh, K. R., Powell, N. E., Norris, A. W., Cotsell, J. N. & Ketola, B. M. The effect of damp-heat and UV aging tests on the optical properties of silicone and EVA encapsulants. *Progress in Photovoltaics: Research and Applications* 19, (2011).
3. Xiu, Y., Zhu, L., Hess, D. & Wong, C. P. Superhydrophobicity and UV stability of polydimethylsiloxane/ polytetrafluoroethylene (PDMS/PTFE) coatings. in *Proceedings of the International Symposium and Exhibition on Advanced Packaging Materials Processes, Properties and Interfaces* vol. 2007 (2007).

Authors' reply: Thank you for providing the valuable references for the modelling and also the PDMS stability. We have cited them in the manuscript.

Reviewer 4:

This manuscript reports a surface textured PDMS film integrating high visible diffusion transmittance, radiative cooling and self-cleaning. This film was fabricated by template method, including photolithography, etching, solution casting and peeling. Although interesting, some of main claims of this work are not well supported by the results. The mechanism of the spectral selective and weatherability properties of the film was not stated. Besides, the self-cleaning performance seems not convinced. Therefore, this version cannot be published yet, unless the author can solve the following issues:

1. One of its principal assertions suggests that micron-scale square-based-pyramid surface structures were devised to enhance solar light transmission, particularly in a diffused manner, improve infrared emissivity within the atmospheric window, and attain superhydrophobicity. Nevertheless, there is an absence of a detailed design foundation, including a lack of comprehensive comparative testing for various geometric shapes and parameters. Although the authors do present two PMMM samples with different widths of 6 and 10 μm , this evidence falls short of substantiating the central novelty of this work.

Authors' reply: Thank you for the constructive comment. We recognize the importance of a detailed investigation into the design foundations and the impact of various geometric parameters on the performance of our PMMM film. In response to the reviewer's concern, we have expanded our experimental scope to include PMMM samples with a wider range of micro-pyramid sizes. We fabricated six samples with pyramid widths of 6 μm , 8 μm , 10 μm , 12 μm , 14 μm , and 16 μm . This approach allows us to explore the effects of these variations more comprehensively. The SEM images of these new PMMM samples are presented in **Fig. 3b** (reproduced below for the reviewer's convenience), demonstrating the precision and uniformity of the pyramid structures across this range of sizes.

Fig 3b. SEM images of PMMM samples with different sizes of the micro-pyramids.

We have performed extensive characterization of these samples, particularly focusing on their optical transmittance and emissivity properties, as detailed in additional **Supplementary Fig. 14** (reproduced below for the reviewer's convenience). The average transmittance in 0.3–2.5 μm and average emissivity in 8–13 μm are summarized in an additional **Fig. 3c** (reproduced below for the reviewer's convenience).

Supplementary Fig. 14. Performance of PMMM samples with various micro-pyramid sizes. **a**, Global transmittance. **b**, Emissivity.

Fig. 3c. Average global transmittance (τ_{g_ave}) in 0.25–2.5 μm , and average emissivity (ϵ_{ave}) in 8–13 μm for PMMM with different pyramid sizes.

The variations in both spectrally-weighted average transmittance (τ_{g_ave}) and average emissivity (ϵ_{ave}) across pyramid sizes ranging from 6 to 16 μm are relatively minor. To facilitate comparison, we calculated the differences in τ_{g_ave} and ϵ_{ave} for pyramid units of different sizes against the 10- μm -wide pyramid units, as illustrated in **Supplementary Fig. 14c**. The observed variations in τ_{g_ave} and ϵ_{ave} are less than 0.014 across this size range.

Supplementary Fig. 14 c. Differences in τ_{g_ave} and ϵ_{ave} for pyramid units of different sizes against the 10- μm -wide units.

We also examined the contact angles for these varying pyramid sizes. To facilitate comparison, we calculated the differences in contact angle θ for pyramid units of different sizes against the 10- μm -wide units, as illustrated in **Supplementary Fig. 14d**. The variation in contact angle is less than 3° across the range of pyramid sizes, translating to a change of less than 2%.

Supplementary Fig. 14 d. Differences in contact angle θ for pyramid units of different sizes against the 10- μm -wide units.

The performance of light diffusion in our PMMM is fundamentally influenced by the base angle of the micro-pyramids. It is important to note that our current fabrication method, which involves etching <100>-oriented Si wafers, restricts us to a fixed base angle of 54.7° , preventing the creation of micro-pyramids with varying base angles. However, to understand the potential effects of different base angles, we engaged in additional analytical and simulation work to provide more evidence. The findings are detailed in **Supplementary Fig. 6**. Through analysis based on Snell's law, we determined that the diffusion angle φ (or "refraction angle") increases with the enlargement of the micro-pyramid's base angle (θ_b). Our simulations demonstrate that φ ranges from 18° to 41° as θ_b varies from 40° to 70° . This range indicates a significant potential for improved sunlight utilization in indoor environments, which could contribute to a more pleasant lighting ambiance. Furthermore, as shown in **Supplementary Figs. 6a**, the value of φ , as per Snell's law, does not depend on the size of the pyramids, but on their base angles.

Supplementary Fig. 6. Correlation between the diffusion angle and the base angle of the micro-pyramid.

Overall, the PMMM's performance exhibits a strong tolerance to variations in pyramid size. As a result, the need for precise control over pyramid dimensions during fabrication is reduced, underscoring the material's robustness and adaptability in manufacturing processes. In demonstrating the diverse functionalities of the PMMM, we chose a sample with $10\text{-}\mu\text{m}$ -wide micro-pyramid units, which allows us to showcase the detailed optical properties in **Fig. 2**, the material's outdoor radiative cooling performance in **Fig. 4**, and its self-cleaning capabilities in **Fig. 5**.

The experiments, simulations, and analyses provide insights into the relationships between pyramid geometry and the multifunctional capabilities of PMMM. This exploration substantiates the central novelty of our work, which lies in the design and evaluation of multifunctional metamaterials that simultaneously address the challenges faced by traditional glass materials through light management, self-cleaning, and radiative cooling properties. We hope that the reviewer will find that these additional findings and analysis provide more detailed guidance and foundation for designing such metamaterials, thereby more clearly substantiating the central novelty of this work.

The additional **Figs. 3b** and **3c**, and **Supplementary Figs. 6** and **14** have been added to the manuscript and supplementary file. The above explanation has been added to the manuscript on Pages 7, 8 and 11.

2. Although the authors conducted Ray-tracing simulations to visually depict the light distribution characteristics of the film, these simulations fail to elucidate the connection between the average transmittance, diffused transmittance, and the textured surface's shape. Additionally, the author's attribution of the heightened emissivity to the gradual alteration of the refractive index induced by the pyramid's shape remains vague; it does not sufficiently clarify how the shape of pyramid impacts the intensity and wavelength enhancements.

Authors' reply: To address the reviewer's concerns, we conducted additional simulations focused on elucidating the correlation between the transmittance properties and the microstructure of the PMMM. Our simulation model assumes wavelength-independent refractive indices for PDMS and soda-lime glass. **Supplementary Fig. 13a** shows the schematics of the global and diffused transmittance modelling. These simulations yield $\tau_{g_ave_sim}$ (global transmittance simulated) of 93.3% and $\tau_{dif_ave_sim}$ (diffused transmittance simulated) of 72.2%. These results align well with our experimental results ($\tau_{g_ave_exp} = 95\% \pm 1\%$ and $\tau_{dif_ave_exp} = 73\% \pm 1\%$). Similarly, the $\tau_{g_ave_sim}$ for the glass substrate is calculated to be 91.8%, comparable to the experimental $\tau_{g_ave_exp}$ of $92\% \pm 1\%$. The simulation results in **Supplementary Fig. 6** shows that the refraction angle is 27.0° , which agrees well with the tested refraction angle of 27.0° .

Supplementary Fig. 13. Schematics of the global and diffused transmittance modelling.

The PMMM demonstrates a higher global transmittance, which is explicable through a comparison with the reflective properties of a planar film. Specifically, rays reflected off a flat film surface typically propagate back towards the light source, leading to the material's reflectance. In contrast, within the PMMM's micro-pyramid structure, rays reflecting off the inclined surfaces of the micro-pyramids are redirected towards adjacent micro-pyramids. This redirection causes the reflected rays to be transmitted deeper into the PMMM, thereby enhancing its transmittance. Consequently, the τ_g for PMMM surpasses that of a standard glass substrate.

Furthermore, the connection between the diffused transmittance (τ_{dif}) and the PMMM's microstructure can be understood as follows: Diffused rays are generated by the refraction and reflection of light at the surfaces of the micro-pyramids. Conversely, incident light rays that strike the flat areas between the micro-pyramids travel straight through the PMMM without contributing to diffusion. Therefore, τ_{dif} is closely linked to the proportion of incident light that interacts with the micro-pyramids relative to the total incident light. The diffused transmittance thus can also be estimated as $(w-g)^2/w^2 = 72.6\%$, which agrees with the ray-tracing simulation result of 72.2%, and the experimental result of $73\% \pm 1\%$. The

connection between the diffusion angle (or “refraction angle”) and the micro-pyramid structure is detailed in the response to the Comment 1 and the additional **Supplementary Fig. 6**.

Supplementary Fig 9. Refractive index gradient of the micro-pyramid structure and absorption mechanism of the PMMM in the MIR range.

For the mechanism behind the enhanced absorption and emissivity properties of PMMM, this enhancement is a result of the gradual change in the refractive index within the micro-pyramid structure. Fan’s group from the Stanford University (*Zhu, ..., Fan., Optica, 1, 32-38, 2014*) firstly used this theory to explain the enhancement of emissivity by silica micro-pyramids. To solve the concern raised by the reviewer, we prepare an additional **Supplementary Figs. 9** to more clearly clarify how the shape of pyramid generates the gradual change in the refractive index, and further impacts the enhancement of absorptance and emissivity. As shown in **Supplementary Figs. 9a and 9b**, at the apex of the micro-pyramid, the volume fraction of air is predominant, leading to a refractive index that closely approximates that of air, as per the effective medium theory. Progressing towards the base of the micro-pyramid, the proportion of PDMS increases, thereby shifting the refractive index closer to that of PDMS. While the targeted wavelength range (8–13 μm) is similar to the size of the pyramids, the refractive index gradient plays a crucial role in reducing the contrast at the air-PDMS interface, thereby diminishing the reflection of mid-infrared radiation that typically occurs at this boundary. Consequently, the micro-pyramid's refractive index gradient is instrumental in minimizing surface reflection as shown in **Supplementary Figs. 9c and 9d**, allowing the PMMM to absorb a significant amount of incoming mid-infrared radiation. Furthermore, according to Kirchhoff's law of thermal radiation, this absorption enhancement directly translates to increased emissivity. Our experimental results indicate that varying the size of the micro-pyramids within a range of 6–16 μm has a minimal effect on the average emissivity in the 8–13 μm range (detailed in the response to Comment 1). This is primarily due to the slight impact of micro-pyramid size on the gradual change in the refractive index when the base angle of the pyramids is fixed. In future research, it is interesting to explore the effects when the micro-pyramid size is significantly larger or smaller than the targeted wavelength range of 8–13 μm .

The above explanation on the connections between the microstructure of PMMM and its optical properties (global average transmittance, diffused transmittance and refraction angle) has been added to Pages 6 and 7 in the manuscript and also the supplementary file. The above further clarification on the enhancement of the emissivity via the micro-pyramids has been added on the Page 6 and also the supplementary file.

3. The authors delivered only a superficial analysis apparent in the assessment of surface wettability. The authors showcased the superiority of the surface with 10 μm -wide micro-pyramids over that of 6 μm -wide micro-pyramids. Nonetheless, the manuscript lacks in-depth analysis and discussion in this regard.

Authors' reply: Thank you for the comment. We have provided SEM images of the two PMMM samples in **Supplementary Figs. 18a** and **18b**. In these images, the width of the micro-pyramid units in both samples is $w = 10 \mu\text{m}$, as indicated by the red dashed rectangular regions in the SEM images. The spaces between adjacent pyramids, denoted as δ , are approximately 4 μm and 1.5 μm , respectively. This means the actual widths of the pyramids are 6 μm and 8.5 μm , not 6 μm and 10 μm as initially mentioned – we acknowledge and correct this error in our original description.

Supplementary Fig. 18. **a**, SEM image of PMMM sample with $\delta = \sim 4 \mu\text{m}$ gap between adjacent pyramids. **b**, SEM image of PMMM sample with $\delta = \sim 1.5 \mu\text{m}$ gap. **c**, Schematic illustrating the inverse relationship between δ and the contact angle, with a flat surface ($\delta \approx 10 \mu\text{m}$) as the extreme condition.

Our additional analysis, as depicted in **Supplementary Fig. 18c**, illustrates that an increase in the spacing (δ) between the micro-pyramids leads to a corresponding increase in the flat area of the PMMM. This larger flat area facilitates the spreading of water droplets on the PMMM surface, thereby reducing the contact angle. For example, the PMMM sample with $\delta = \sim 1.5 \mu\text{m}$ exhibits a higher contact angle of 152° compared to the 128° of the sample with $\delta = \sim 4 \mu\text{m}$. This behavior aligns with the principles of the *Wenzel* equation: $\cos\theta = r \cdot \cos\theta_0$, where θ represents the contact angle on the structured surface, θ_0 is the contact angle on the flat base material, and r is the roughness ratio (the ratio of actual to projected solid surface area). As δ increases, the roughness ratio r decreases, leading to a reduction in the contact angle θ .

We have added this detailed explanation to Page 11 of the manuscript and the supplementary file for a more comprehensive understanding of the relationship between pyramid spacing and contact angle variations on the PMMM surface.

4. The authors stated that the superhydrophobic surface can effectively achieve self-cleaning performance, whether through rain in an active manner or dew passively. While it's easy to comprehend how droplets can remove dust, the mechanism for dust removal by dew, as illustrated in Figure 4F, remains unconvincing. I believe that the dust might accumulate in cavities that droplets or

dew may not reach, especially in the Cassie model. To bolster this claim, the authors should consider conducting a comparison between the film and glass surfaces.

Authors' reply: Thank you for the insightful comment. We acknowledge the reviewer's point about the potential for smaller dust particles (e.g., less than 5 μm) to become lodged in the micro cavities, which aligns with findings from our group's prior research, as documented in Roslizar *et al.* (*Solar Energy Materials and Solar Cells*, 214, 110582, 2020).. To address the reviewer's concern, we have analyzed the dust particles used in this study (the real dust particles were collected from a local window in Karlsruhe, Germany), as detailed in **Supplementary Fig. 19** (reproduced below for the reviewer's convenience). SEM imaging reveals that the majority of dust particles in Karlsruhe are larger than 10 μm . Consequently, these particles are unlikely to be trapped in the micro cavities on the PMMM surface. This aligns with the observations in the **Supplementary Video 2** and **3**, and **Fig. 4f**, where dew effectively removes most dust particles from the surface.

Supplementary Fig. 19. SEM/EDX characterization

According to the reviewer's suggestion, we also conducted comparative experiments on a glass surface. As demonstrated in **Supplementary Fig. 20a**, the glass surface, being hydrophilic, struggles with active cleaning by water droplets. We used the same amount of water droplets to clean the glass surface as we used to clean the PMMM samples. The droplets tend to adhere to the glass, making it difficult to effectively dislodge dust with the limited amount of water droplets. More water is required to remove the dust further and fully on the glass surface. For the dew cleaning as shown in **Supplementary Fig. 20b**, the hydrophilic nature of the glass surface leads to the formation of a dew film instead of discrete droplets, lacking the roll-off effect necessary for effective cleaning. As a result, the wetted dust remains

adhered to the glass surface, making it more challenging to remove effectively at the end of the dew formation process. In contrast, the hydrophobic PMMM surface easily forms noticeable dew droplets that can roll off, sweeping across the surface and efficiently removing dust.

Supplementary Fig. 20. Cleaning processes of a glass plate. **a**, Active cleaning process by water droplets (indoor lab simulated experiment). **b**, passive cleaning process by dew (indoor lab simulated experiment).

The above explanation of the cleaning process has been added to Page 13 of the manuscript and add the supplementary figure to the supplementary file.

5. The film thickness seems to exhibit a distinct inconsistency within this study. For instance, in the simulation section, the authors specified the film thickness as 50 μm , whereas Figure 2D depicts the film with a thickness of 70 μm .

Authors' reply: Thank you for highlighting this discrepancy. In response, we have rigorously updated our simulations to reflect a consistent film thickness of 70 μm across all models (this matching the experimental data). Additionally, we have adjusted the size of the gaps between the micro-pyramids to align with the results obtained from our SEM measurements.

Consequently, we have revised all relevant figures, including **Fig 3a** and **Supplementary Figs. 10, 11 and 12** to reflect these changes. It is important to note that these updates do not alter the core findings of our study. However, we acknowledge and appreciate the reviewer's input, as it has undeniably enhanced the accuracy and quality of our simulations.

REVIEWER COMMENTS

Reviewer #1 (Remarks to the Author):

After the corrections already made, it can be published.

Reviewer #2 (Remarks to the Author):

All the comments raised by this reviewer have been satisfactorily addressed. Thank you for your great efforts.

Reviewer #3 (Remarks to the Author):

In this revision of their manuscript "Radiative cooling, light management and self-cleaning based on polymer metamaterials," the authors performed a series of new outdoor experiments demonstrating the PMMM's efficacy as a passive radiative cooler and expanded greatly on the self-cleaning nature of their polymer-based micro-photonic multi-functional metamaterial (PMMM). These improvements, in my opinion, make the work even more interesting to readers of Nature Communications. Further, I believe the authors have satisfied the concerns raised in my previous comments, as well as those raised by other reviews, with the exception of one issue. I will detail it below.

The authors did not initially provide an uncertainty budget for their spectrophotometric measurements. After requesting one, the authors have now provided a measure of repeatability of their measurements. This is insufficient. Repeatability only captures Type A components of an uncertainty budget. By missing every type B component, the authors' claimed uncertainties exceeded (especially in the infrared) the capabilities of best-in-the-world National Metrology Institutes, such as Physikalisch-Technische Bundesanstalt (PTB) in Germany, the National Institute of Standards and Technology (NIST) in the United States, and National Physical Laboratory (NPL) in the United Kingdom. The authors must complete a careful analysis of the Type B uncertainty components to their measurements and present an accurate statement of their measurement uncertainty. I would direct the authors to review works from these three laboratories, all of which have mature spectrophotometry capabilities. As a starting point, I can direct the authors to NIST Special Publication 250-94 (<http://dx.doi.org/10.6028/NIST.SP.250-94>) for one approach to this problem. While it is certainly not the only way to approach the problem, it is very thorough and contains several references as appendices which contain uncertainty budgets from NIST and NPL.

Once this issue is resolved, I recommend this work for publication.

Reviewer #4 (Remarks to the Author):

Comments:

The authors have addressed almost the majority of my comments. However, given the potential for resonance phenomena to occur when the pyramid size is comparable to the incident light, the current ray-tracing simulation method may not adequately capture this aspect. To elucidate the influence of the pyramid structure on its infrared emissivity, it is strongly recommended to employ a simulation method capable of representing such resonances.

I would like to bring to the authors' attention a recent study that utilized the Finite-Difference

Time-Domain (FDTD) simulations to investigate the impact of surface structure on emissivity (Ref: Adv. Funct. Mater. 2023, 2305650). Integrating insights from this work could explain the infrared emissivity disparity with different structures (Fig.3c, Supplementary Fig. 14.) and further enhance the soundness of the research reported in the manuscript. Following the incorporation of this improvement, the acceptance of the manuscript may be attainable.

Radiative cooling, light management and self-cleaning based on polymer metamaterials

Gan Huang^{1,*}, Ashok R. Yengannagari¹, Kishin Matsumori¹, Prit Patel¹, Anurag Datla¹, Karina Trindade¹, Enkhlen Amarsanaa¹, Tonghan Zhao¹, Uwe Köhler¹, Dmitry Busko¹, Bryce S. Richards^{1,2,*}

Response to Reviewers

Reviewer #3:

In this revision of their manuscript “Radiative cooling, light management and self-cleaning based on polymer metamaterials,” the authors performed a series of new outdoor experiments demonstrating the PMMM’s efficacy as a passive radiative cooler and expanded greatly on the self-cleaning nature of their polymer-based micro-photonic multi-functional metamaterial (PMMM). These improvements, in my opinion, make the work even more interesting to readers of Nature Communications. Further, I believe the authors have satisfied the concerns raised in my previous comments, as well as those raised by other reviews, with the exception of one issue. I will detail it below.

The authors did not initially provide an uncertainty budget for their spectrophotometric measurements. After requesting one, the authors have now provided a measure of repeatability of their measurements. This is insufficient. Repeatability only captures Type A components of an uncertainty budget. By missing every type B component, the authors’ claimed uncertainties exceeded (especially in the infrared) the capabilities of best-in-the-world National Metrology Institutes, such as Physikalisch-Technische Bundesanstalt (PTB) in Germany, the National Institute of Standards and Technology (NIST) in the United States, and National Physical Laboratory (NPL) in the United Kingdom. The authors must complete a careful analysis of the Type B uncertainty components to their measurements and present an accurate statement of their measurement uncertainty. I would direct the authors to review works from these three laboratories, all of which have mature spectrophotometry capabilities. As a starting point, I can direct the authors to NIST Special Publication 250-94 (<http://dx.doi.org/10.6028/NIST.SP.250-94>) for one approach to this problem. While it is certainly not the only way to approach the problem, it is very thorough and contains several references as appendices which contain uncertainty budgets from NIST and NPL.

Once this issue is resolved, I recommend this work for publication.

Authors’ reply: We appreciate your constructive feedback, particularly regarding the comprehensive uncertainty analysis of our spectrophotometric measurements. As per your recommendation, we have undertaken a detailed examination of both Type A and Type B uncertainties, using NIST Special Publication, among other references¹, to refine our analysis. Indeed, the total uncertainty was higher than we originally calculated, and especially for the FTIR measurements.

The updated uncertainties have been added on Page 6 of the manuscript and further detailed in Supplementary Fig. 7, now includes a more comprehensive error propagation for all measured values. Our findings reveal that the total uncertainty for FTIR measurements, considering both random errors and systematic uncertainties, is higher than initially estimated, mainly because of the uncertainty of reference standards reflectivity. Specifically, the uncertainty ranges for the averaged transmission and

emissivity values have been adjusted to 0.4–2.5 rel.% and 5.5 rel.%, respectively. These adjustments reflect a more realistic understanding of our measurement capabilities.

The details have been added in the supplementary file (under **Supplementary Fig. 7**), as reproduced below:

The measurements of the reflection and transmission properties were carried out using UV-Vis-NIR Spectrophotometer (Agilent Cary 7000) across a wide range of wavelengths (300–2500 nm). The instrument was equipped with an integrated sphere with a highly reflective inner surface to collect both diffused and direct light. All measurements were done using a NIST-calibrated diffuse reflectance standard. The emissivity was measured by a Fourier-transform infrared spectroscopy (Bruker Vertex 70) which was equipped with an integrated sphere (A562) with a highly infrared reflective inner surface coated with gold. The equations used to calculate the values of τ_{g_ave} , τ_{dif_ave} , τ_{vis_ave} and ϵ_{ave} can be found in the Method section.

Corresponding uncertainties for values above were calculated according to a published guide [Woolliams, Emma R. "Determining the uncertainty associated with integrals of spectral quantities." (2013), Technical Report. EMRP Joint Research Project]. Random errors impact was estimated using acquired transmission/reflection spectra and 10-times repetitive measurements with a repositioned sample, whereas the systematic uncertainties (type B) were valued with best knowledge using a technical specification of both Cary 7000 (DRA attachment) and Bruker Vertex 70. All systematic uncertainties were quadratically summed and added to random errors to get a total uncertainty of the measurement.

For the measurements in UV/Vis/NIR spectra, the random error and all systematic uncertainties caused by the nonideality of the system are quite low (0.1–2 rel.%), depending on an acquired value (for ~100% and ~10% transmission/reflection, accordingly). The total uncertainty is mainly governed by an uncertainty of reflectivity estimation of the calibrated standard (~0.5% in UV/Vis/NIR ranges 0.3–2 μm and ~3% in the NIR range (>2 μm)) or an estimated bias of the uncalibrated reference golden port in FTIR measurements. Of note is that as the average transmission in our calculation is weighed by the AM1.5 solar spectrum, the main intensity of which is concentrated in the visible range, a moderately high uncertainty of the standard in NIR range has only a minor impact on the resulting total uncertainty. In the FTIR range, both random noise and estimated systematic uncertainties are relatively higher and are about 2–5 rel.%. It is worth to note that in a mutual comparison between glass and PMMM systems, the main uncertainty caused by a reflectivity of the reference port close to 100% but having unknown absolute value is equally applied to glass and PMMM samples. Summary of calculated integrals with uncertainties: (1) Glass sample: $\tau_{g_ave} = 0.906 \pm 0.004$; $\tau_{dif_ave} = 0.014 \pm 0.0003$; $\tau_{vis_ave} = 0.909 \pm 0.004$; $\epsilon_{ave} = 0.87 \pm 0.05$; (2) PMMM sample: $\tau_{g_ave} = 0.949 \pm 0.004$; $\tau_{dif_ave} = 0.731 \pm 0.003$; $\tau_{vis_ave} = 0.952 \pm 0.025$; $\epsilon_{ave} = 0.98 \pm 0.05$.

Type A and B uncertainties are summarized in the new **Supplementary Table 1**, as reproduced below:

Supplementary Table 1. Type A and B uncertainties for Agilent Cary 7000 DRA and Bruker Vertex 70 acquired using their specification and acceptance tests.

Uncertainties art	Type A relative uncertainties
Total sample position repeatability UV/Vis/NIR	0.004
Total sample position repeatability FTIR	0.001

Random noise	Calculated using experimental data					
	Type B relative uncertainties					
	Uncertainty value in range for Cary 7000					FTIR Vertex 70
	250–600nm	601nm–860nm	860nm–1500nm	1.5–2.2µm	2.2–2.5µm	2.5–16µm
Corrected baseline flatness (lamp stability)	0.0016	0.0016	0.0016	0.0016	0.0016	
Photometric Noise (RMS)	0.00012	0.00012	0.0001	0.0001	0.0001	
Photometric linearity	0.0007	0.0007	0.0015	0.0015	0.0015	
Photometric accuracy						0.001
Stray light	0.0001	0.0001	0.0001	0.0001	0.0001	
Beam polarisation	0.001	0.001	0.001	0.001	0.001	
reference standard reflection (k=2), 95% confidence	0.0053	0.0049	0.0049	0.0088	0.032	
reference port reflection value bias						0.05
Total relative error Type B (sqrt of quadrature sum) in ordinate axis	0.0057	0.0053	0.0055	0.0091	0.0321	0.05
Uncertainty in an abscissa (wavelength) axis	0.008 nm	0.008 nm	0.4nm	0.4 nm	0.4 nm	0.4cm⁻¹

Additional reference:

1. Woolliams, Emma R. "Determining the uncertainty associated with integrals of spectral quantities." (2013), Technical Report. EMRP Joint Research Project.

Reviewer #4:

The authors have addressed almost the majority of my comments. However, given the potential for resonance phenomena to occur when the pyramid size is comparable to the incident light, the current ray-tracing simulation method may not adequately capture this aspect. To elucidate the influence of the pyramid structure on its infrared emissivity, it is strongly recommended to employ a simulation method capable of representing such resonances.

I would like to bring to the authors' attention a recent study that utilized the Finite-Difference Time-Domain (FDTD) simulations to investigate the impact of surface structure on emissivity (Ref: Adv. Funct. Mater. 2023, 2305650). Integrating insights from this work could explain the infrared emissivity disparity with different structures (Fig.3c, Supplementary Fig. 14.) and further enhance the soundness of the research reported in the manuscript.

Following the incorporation of this improvement, the acceptance of the manuscript may be attainable.

Authors' reply: We appreciate your important comment to deepen our understanding of the PMMM and improve the discussion about its optical properties. We agree with your point that the micro-pyramid structures may have resonance modes. To understand the resonance modes of the micro-pyramid structures, we employed Cartesian multipole decomposition (CMD) using the wave-optics module of COMSOL Multiphysics, which is based on the finite element method. According to our new simulation results, we confirmed that the micro-pyramid structures in this study possess resonance modes. Thus, the refractive index gradient cannot properly describe the absorption mechanism of the PMMM. We have thus deleted the refractive index gradient in the manuscript, and replaced the previous Supplementary Fig. 9 with a new figure (reproduced below for the convenience of the reviewer), which shows the absorption/emissivity enhancement mechanism. We also appreciate the reviewer providing the reference (Adv. Funct. Mater. 2023, 2305650) about the impact of surface structure on emissivity, which has been cited in the manuscript as Ref. 59.

In general, the optical resonance of a subwavelength dielectric particle is dominated by a (electric or magnetic) dipole resonance. The dipole moment creates scattered waves symmetrically in forward and backward directions. In contrast, a dielectric particle whose size is comparable to or larger than a wavelength supports different order resonance modes, resulting in strong forward scattering. When a metasurface comprises small or large particles, such a metasurface has strong reflection and transmission, respectively. The resonance mode analysis based on CMD suggests that the small micro-pyramid unit ($w = 6 \mu\text{m}$) mainly support an electric dipole resonance, and the large micro-pyramid unit ($w = 16 \mu\text{m}$) has a complex resonance by a mixture of different order modes. Even though the micro-pyramid's resonance state differs depending on its size, we could not observe such an optical response difference for the PMMM samples with different pyramid sizes as shown in the Fig. 3c, and Supplementary Fig. 14.

To understand this phenomenon, we observed the electric field distribution maps as shown in the new Supplementary Fig. 9. We found that the incident wave couples with the resonance modes of the micro-pyramid. The incident wave, which is confined by the resonance modes, then efficiently couples to the PDMS film. This is attributed to the fact that the bottom of the micro-pyramid structure contacts the flat PDMS film, which has a refractive index higher than air. The coupling of the incident wave to the PDMS film through the micro-pyramid's resonance modes results in transmission enhancement at the PDMS-air interface. The small micro-pyramid achieves strong transmission at the interface because of this coupling effect. The large micro-pyramid strengthens the transmission

at the interface mainly by its forward scattering. This is because since the large micro-pyramid possesses a complex resonance state, it can have strong forward scattering without the coupling effect. Thus, the mechanism of the transmission enhancement at the interface differs for different micro-pyramid sizes. However, the PMMM can possess strong transmission at its interface regardless of the micro-pyramid's size, resulting in the micro-pyramid size-independent high emissivity of the PMMM in MIR range.

We have made revision in the main text on Pages 6 and 7 to explain the emissivity enhancement mechanism in the MIR range based on the resonance mode analysis:

“In previous works, micro-pyramid structures have been utilized to allow the system's refractive index to gradually change from the refractive index of air to the refractive index of polydimethylsiloxane^{1, 56}. This interpretation based on the refractive index gradient can be applied for subwavelength micro-pyramids. However, the micro-pyramid size of the PMMM is comparable to the wavelengths in the MIR range. In this case, resonance modes of the micro-pyramid must be considered to understand the PMMM's absorption mechanism, which can be done by utilizing Cartesian multipole decomposition (CMD)²⁻⁴. According to our resonance mode analysis based on CMD (Supplementary Fig. 9), it is found that the micro-pyramid with $w = 10 \mu\text{m}$ supports a strong electric dipole resonance, and weak magnetic dipole, electric quadrupole, and magnetic quadrupole resonances. Since the micro-pyramid is placed on the polydimethylsiloxane film, those resonance modes can couple to the polydimethylsiloxane film, which can cause strong forward scattering into the polydimethylsiloxane film⁴⁻⁷. Consequently, the reflection at the air-polydimethylsiloxane interface diminishes, more energy of the incident wave can be transmitted into the polydimethylsiloxane film, and the PMMM' absorption in MIR range is thus enhanced based on Beer's law.”

We have also made revision in the main text on Page 8 to explain the influence of the pyramid structure on its infrared emissivity:

“To investigate this effect, we used the wave-optics module in COMSOL Multiphysics to study the electric field distribution around the micro-pyramid at its resonance wavelength⁷, as detailed in Supplementary Fig. 9. We discovered that the incoming wave interacts with the resonance modes of the micro-pyramid structure, leading to its confinement and efficient transfer to the polydimethylsiloxane film. This happens because the micro-pyramid's base directly contacts the polydimethylsiloxane film, which has a higher refractive index than air, facilitating the wave's coupling to the film^{5, 7}. Smaller micro-pyramids enhance transmission effectively through this coupling mechanism. Larger micro-pyramids primarily use their forward scattering to strengthen transmission due to their complex resonance states¹⁰. Therefore, the material can achieve high transmission across its interface for any micro-pyramid size in this study, ensuring size-independent high emissivity in the MIR range.”

More details of the additional modelling have been added to the supplementary file (new Supplementary Fig. 9):

Supplementary Fig. 9. Absorption enhancement mechanism of the PMMM in the MIR range. *a*, Simulation model of the PMMM. Wave-optics simulations were conducted using COMSOL Multiphysics (the finite element method). The model comprises a PDMS micro-pyramid, PDMS film, and soda-lime substrate. The top micro-pyramid has the width and height of w_{MP} and h_{MP} , respectively. The height is given by $h_{MP} = (w_{MP}/2) \times \tan(54.7^\circ)$, where 54.7° is the base angle of the fabricated micro-pyramid. To reduce computational cost to simulate the thick film and substrate, the transition boundary conditions were applied on their bottom surfaces. The perfectly matched layers (PMLs) were set on top and bottom of the calculation domain to consider infinitely large air space. The simulation model was used as a unit cell, and the periodic boundary conditions were applied in the x - and y - directions. The periodicity of the unit cell was given as $w = w_{MP} + g$, where g is the gap between the micro-pyramids. In the simulation, the x -polarized incident wave propagated along the $-z$ direction. *b*, Comparison of different micro-pyramids' optical responses in the atmospheric window range of 8–13 μm . The solid and dashed lines are for absorptance and reflectance, respectively. The reflectance and transmittance of the simulation model were calculated, and absorptance was obtained by $1 - (\text{reflectance}) - (\text{transmittance})$. The micro-pyramid structures labeled as $w = 6 \mu\text{m}$, $10 \mu\text{m}$, and $16 \mu\text{m}$ in Fig. 3b of the main text were simulated. In the simulation, the parameter sets of $(w_{MP}, g) = (4.4, 1.6) \mu\text{m}$, $(8.40, 1.46) \mu\text{m}$, and $(13.55, 1.53) \mu\text{m}$ were used for those micro-pyramids. The parameter sets were obtained from the SEM images. The optical response of a flat surface (without micro-pyramid) was also simulated. Compared with the flat surface and the textured surfaces, it is found that the textured surfaces have higher absorptance, especially around

the wavelength ranges of 9–10 μm and 12–13 μm (blue and yellow shaded areas). The absorption is enhanced because the reflection is weakened at the air-PDMS interface by the micro-pyramid structure. **c**, Resonance mode analysis. The resonance modes of the micro-pyramids were extracted by using the simulation model illustrated in (a) and employing Cartesian multipole decomposition (CMD)². In general, CMD is used for single particles⁸, but it can also be applied to periodic systems³. From CMD, four resonance modes were obtained: electric dipole (ED), magnetic dipole (MD), electric quadrupole (EQ), and magnetic quadrupole (MQ) moments. In the plot, those resonance modes are characterized as scattering efficiencies. The total scattering spectrum of the micro-pyramids is given by the sum of the scattering spectra of those modes. For $w_{\text{MP}} = 4.4 \mu\text{m}$, the total scattering is dominated by the ED resonance. With increasing w_{MP} , the higher-order modes contribute to the total scattering. **d**, Electric field distribution of the single micro-pyramid placed on a substrate with a refractive index of n_{sub} . The maps present the absolute value of the electric field (E) normalized by the electric field of the incident wave (E_0) at the wavelength of 10 μm . It must be mentioned that a simulation model different from the one shown in (a) was used to simulate the single micro-pyramid. For this simulation, the periodic boundary conditions were not applied, but the PMLs were set around the micro-pyramid. For $n_{\text{sub}} = 1$ (air), the micro-pyramid with $w_{\text{MP}} = 4.4 \mu\text{m}$ shows the dipolar-like electric field distribution, where the strong electric near fields are created around the bottom apexes. When n_{sub} increases to 1.5, the near fields weaken since the created electric fields couple into the substrate, resulting in the incident light being strongly scattered into the substrate⁴⁻⁷. For the larger micro-pyramids ($w_{\text{MP}} = 8.40$ and $13.55 \mu\text{m}$) with $n_{\text{sub}} = 1$, their near-field distributions are more complex compared with the smallest micro-pyramid because of their higher-order modes. In a similar manner to the smallest micro-pyramid, the near fields of the larger micro-pyramids become weak when $n_{\text{sub}} = 1.5$. **e**, Forward to backward scattering (F/B) ratio. The F/B ratio was calculated using the same simulation model for (d) to understand the scattering properties of the single micro-pyramid quantitatively. The forward and backward scatterings were defined by the powers of the scattered waves below and above the substrate, respectively. The F/B ratio is plotted as a function of w_{MP} . The blue and red marks are for $n_{\text{sub}} = 1$ and 1.5, respectively. For $w_{\text{MP}} = 4.4 \mu\text{m}$ and $n_{\text{sub}} = 1$, the F/B ratio is about 1.4, meaning that the smallest micro-pyramid scatters the incident wave almost symmetrically into the forward and backward directions because of its ED resonance^{9,10}. When the smallest micro-pyramid is located on the substrate with $n_{\text{sub}} = 1.5$, the F/B ratio is increased to about 4, which proves that the forward scattering becomes strong for a high n_{sub} ⁴⁻⁷. For $w_{\text{MP}} = 8.4 \mu\text{m}$, similarly to the case of $w_{\text{MP}} = 4.4 \mu\text{m}$, the F/B ratio can be significantly improved by increasing n_{sub} . When w_{MP} is increased to 13.55 μm , the F/B ratio is high regardless of whether there is a substrate on the micro-pyramid's bottom, since the largest micro-pyramid possesses strong forward scattering by its complex resonance state¹⁰⁻¹².

Additional references

1. Lee, E., & Luo, T. (2019). Black body-like radiative cooling for flexible thin-film solar cells. *Solar Energy Materials and Solar Cells*, 194, 222-228.
2. Alaei, R., Rockstuhl, C., & Fernandez-Corbaton, I. (2018). An electromagnetic multipole expansion beyond the long-wavelength approximation. *Optics Communications*, 407, 17-21.
3. Butakov, N. A., & Schuller, J. A. (2016). Designing multipolar resonances in dielectric metamaterials. *Scientific Reports*, 6(1), 38487.

4. Balezin, M., Baryshnikova, K. V., Kapitanova, P., & Evlyukhin, A. B. (2018). Electromagnetic properties of the Great Pyramid: First multipole resonances and energy concentration. *Journal of Applied Physics*, 124(3).
5. Van de Groep, J., & Polman, A. (2013). Designing dielectric resonators on substrates: Combining magnetic and electric resonances. *Optics Express*, 21(22), 26285-26302.
6. Spinelli, P., Verschuuren, M. A., & Polman, A. (2012). Broadband omnidirectional antireflection coating based on subwavelength surface Mie resonators. *Nature Communications*, 3(1), 692.
7. Chen, G., Fu, H., Zou, Y., Wang, S., Gao, Y., Yue, T., ... & Zhou, Y. (2023). A Promising Radiation Thermal Protection Coating Based on Lamellar Porous Ca - Cr co - Doped Y3NbO7 Ceramic. *Advanced Functional Materials*, 33(47), 2305650.
8. Terekhov, P. D., Baryshnikova, K. V., Artemyev, Y. A., Karabchevsky, A., Shalin, A. S., & Evlyukhin, A. B. (2017). Multipolar response of nonspherical silicon nanoparticles in the visible and near-infrared spectral ranges. *Physical Review B*, 96(3), 035443.
9. Alae, R., Albooyeh, M., & Rockstuhl, C. (2017). Theory of metasurface based perfect absorbers. *Journal of Physics D: Applied Physics*, 50(50), 503002.
10. Liu, W., & Kivshar, Y. S. (2018). Generalized Kerker effects in nanophotonics and meta-optics. *Optics Express*, 26(10), 13085-13105.
11. Alae, R., Filter, R., Lehr, D., Lederer, F., & Rockstuhl, C. (2015). A generalized Kerker condition for highly directive nanoantennas. *Optics Letters*, 40(11), 2645-2648.
12. Dezert, R., Richetti, P., & Baron, A. (2019). Complete multipolar description of reflection and transmission across a metasurface for perfect absorption of light. *Optics Express*, 27(19), 26317-26330.

REVIEWERS' COMMENTS

Reviewer #3 (Remarks to the Author):

The authors have satisfied all of my concerns with this round of revisions. I commend the authors for the time and effort they put into creating a comprehensive uncertainty budget for their measurement and recommend this excellent scholarship for publication in Nature Communications.

Reviewer #4 (Remarks to the Author):

The authors have addressed all my concerns raised in the comments, and I believe the manuscript is now acceptable for publication in the journal of Nature Communications.